

# Simulations and observation of nonlinear waves on the continental shelf: KdV solutions

Kieran O'Driscoll[1], Murray Levine*

[1]Department of Civil Engineering, Queen's University Belfast, Belfast, BT9 1NN, Northern Ireland

*deceased

*Correspondence to*: Kieran O'Driscoll (kieran.odriscoll@qub.ac.uk)

**Abstract.** Numerical solutions of the Korteweg-de Vries (KdV) and extended Korteweg-de Vries (eKdV) equations are used to model the transformation of a sinusoidal internal tide as it propagates across the continental shelf. The ocean is idealized as being a two-layer fluid, justified by the fact that most of the

oceanic internal wave signal is contained in the gravest mode. The model accounts for nonlinear and dispersive effects but neglects friction, rotation, and mean shear. The KdV model is run for a variety of idealized stratifications and unique realistic topographies to study the role of the nonlinear and dispersive effects. In all model solutions the internal tide steepens forming a sharp front from which a packet of nonlinear solitary-like waves evolves. Comparisons between KdV and eKdV solutions is explored. The

model results for realistic topography and stratification are compared with observations made at moorings off Massachusetts in the Middle Atlantic Bight. Some features of the observations compare well with the model. The leading face of the internal tide steepens to form a shock like front, while nonlinear high frequency waves evolve shortly after the appearance of the jump. Although not rank ordered, the wave of maximum amplitude is always close to the jump. Some features of the observations are not found in the

model. Nonlinear waves can be very widely spaced and persist over a tidal period.



## 1. Introduction

Internal waves are present throughout earth's oceans wherever there is stratification, from the shallowest near-shore waters to the deepest seas. Internal waves are important to physical oceanographers because they transport momentum and energy, horizontally and vertically, through the ocean, e.g. Munk (1981), Gill (1982). They provide shear to turbulence which results in energy dissipation and vertical mixing, e.g. Holloway (1984), Sandstrom & Elliott (1984). Biological oceanographers are interested because the internal waves carry nutrients onto the continental shelf and into the euphotic zone, e.g. Shea & Broenkow (1988), Sandstrom & Elliott (1984), and Holloway et al. (1985). They are of interest to geological oceanographers because the waves produce sediment transport on the shelf, e.g. Cacchione & Drake (1986). Civil, hydraulic and ocean engineers are interested because the internal waves generate local tidal and residual currents, e.g. Willmott & Edwards (1987), which can cause scour on nearshore as well as offshore structures, e.g. Osborne et al. (1978). Large nonlinear IWs are also of interest to the navy because they cause large vertical displacements and large vertical velocities that may affect underwater operations.

This study is focused on the internal tide and subsequent evolution of nonlinear waves. Internal waves in the ocean span the frequency spectrum from the buoyancy frequency, $N$, to the inertial frequency, $f$. However, the internal, or baroclinic, tide accounts for a large fraction of the energy contained in these waves. The internal tide is generated by the interaction of barotropic tidal current with topography and not directly by the gravitational attraction of sun and moon. The properties and propagation of linear internal tide and waves have been treated in detail by many investigators, see, for example, Garrett & Munk (1979), or the monographs by Gill (1982), Lighthill (1978), or Apel (1987).

As the internal tide shoals, the nonlinear terms in the Navier-Stokes equations become important. These tidal waves of finite amplitude may evolve into packets of high frequency nonlinear waves. The equations describing these waves are much more complex than the linear equations and few mathematical solutions have been found.

5     We are interested in nonlinear internal waves because they are a very energetic part of the signal in time series that we have observed on continental shelves and in the shallow ocean. We are guided by numerical solutions of Korteweg-de Vries (KdV) type equations that incorporate both weak nonlinear and weak dispersive effects.

The state of the art on the evolution of internal solitary waves across the continental shelf is reviewed in Grimshaw et al. (2010). Grimshaw et al. (2004) simulated the transformation of internal solitary waves across the North West shelf of Australia, the Malin shelf edge, the Arctic shelf; Holloway (1987) discussed the evolution of the internal tide in a two-layer ocean on the Australian North West Shelf. Our model simulations of the evolution of the internal tide across realistic in the Middle Atlantic Bight topography cases are unique since these waves have never been modelled across such topography and stratifications, but the model results are compared with observations made at moorings off Massachusetts during the Coastal Mixing and Optics Experiment.

The goal of this paper is to study the observed variability in the evolution of the internal tide as it crosses the continental shelf resulting from different stratifications and varying topography.

The theoretical background of the Korteweg-de Vries (KdV) equation and an extended form of it, the eKdV, are presented in section 2. In section 3, the model framework is presented, and model runs and results of simulations are discussed. Model results are compared with data and observations collected at



the site of the Coastal Mixing and Optics experiment (CMO) in section 4. A summary and conclusions

are presented in section 5.

## 2. Theoretical Background

The Korteweg de Vries (KdV) equation is well known to be a suitable physical model for

describing weakly nonlinear advective effects and linear dispersion in internal waves. It was originally

developed by Benney (1966) and extended to second order by Lee & Beardsley (1974). The KdV equation

is derived from classical nonlinear long wave theory using a two-parameter perturbation expansion in $\varepsilon$

and $\delta$ which scale the nonlinear and dispersive effects, respectively.

The KdV equation, derived following the procedure of Lee & Beardsley (1974) and the discussion

by Lamb & Yan (1996), but without mean current, is given by

$$\eta_t + c\eta_x + \alpha\eta\eta_x + \beta\eta_{xxx} = 0 \qquad\qquad 1$$

where $\eta$ is the vertical displacement amplitude of the wave mode, $c$ is the linear long wave phase speed

for the mode whose amplitude is $\eta$, $\alpha$ and $\beta$ are coefficients of the non-linear and dispersive terms, while

subscripts represent derivatives in time, $t$, and space, $x$, respectively.

     Progressing to $2^{nd}$ order in $\varepsilon$ and $\delta$ (nonlinear and dispersive effects) yields four additional terms

to Eq.(1) - a cubic nonlinear term, as well as higher-order linear and nonlinear dispersive terms - and is

known as the fully extended KdV equation (feKdV). Often only the second-order nonlinear term ('cubic

nonlinearity') is added resulting in the extended KdV (eKdV) equation

$$\eta_t + c\eta_x + (\alpha + \alpha_1\eta)\eta\eta_x + \beta\eta_{xxx} = 0 \qquad\qquad 2$$



where $\alpha_1$ is the coefficient of the cubic nonlinear term. In a two-layer model, for example, when the layers are of similar depth, or when the quadratic nonlinear term is small, the higher-order linear and nonlinear dispersive terms can be omitted, see discussion in Grimshaw et al. (2002). Continuous stratification can support an infinite number of modes. For simplicity we consider wave propagation in a two-layer

stratification which supports one mode only. The justification for making this approximation is that most of the energy in the ocean appears to be contained in the first mode anyway, e.g. Alford & Zhao (2007) and discussion therein, while the shelf often has the appearance of a two-layer stratification: an upper mixed layer separated from a weakly stratified bottom layer by a thin pycnocline. This approximation greatly simplifies the problem; the numerical scheme is much less complex for the two-layer case than

the continuously stratified case, and the results are easier to interpret. The coefficients of the KdV and eKdV equations are greatly simplified for a two-layer fluid and are written (e.g. Ostrovsky & Stepanyants, 1989)

$$c = \sqrt{\frac{g\Delta\rho}{\rho}\frac{h_1 h_2}{h_1 + h_2}} \; ; \qquad \alpha = \frac{3}{2}c\frac{h_1 - h_2}{h_1 h_2} \; ; \qquad\qquad \text{3(a,b)}$$

$$\beta = c\frac{h_1 h_2}{6} \; ; \qquad \alpha_1 = -\frac{3c}{8h_1^2 h_2^2}\left(h_1^2 + h_2^2 + 6h_1 h_2\right) \qquad\qquad \text{3(c,d)}$$

where $\Delta\rho$ is the density difference between upper and lower layers, and $h_1$, $h_2$ are thicknesses of the two layers. We are interested in applying the KdV and eKdV equations to conditions of spatially varying coefficients. This problem has been investigated for slowly varying topography and stratification by Grimshaw (1979) and Pelinovsky et al. (1977). The eKdV equation then has variable coefficients and an additional term:




$$\eta_t + c\eta_x + (\alpha + \alpha_1\eta)\eta\eta_x + \beta\eta_{xxx} + \frac{c}{2Q}Q_x\eta = 0 \qquad (4)$$

where $\qquad Q = \dfrac{Mc^3}{M_0 c_0^{\ 3}} \quad$ and $\qquad M = \dfrac{h_1 + h_2}{h_1 h_2} \qquad$ 5(a,b)

and both $c$ and $M$ in $Q$ vary in the horizontal direction, where the zero subscript indicates a constant value

at a predetermined position. We note that the effective depth, $h'$, is the inverse of the parameter $M$ , Eq.

(5b), contained in $Q$. For level bottom with horizontal interface the values of $h_1$ and $h_2$ are constant and

$Q=1$ everywhere so the horizontal variability term vanishes, and the canonical KdV is the valid model.

The variable coefficient KdV equation is the same as the variable coefficient eKdV equation but with $\alpha_1$

$= 0$. For convenience in solving the equation, we avail of a transformation, utilized by Pelinovsky &

Shavratsky (1976), of the space and time variables $x$ and $t$ to variables $l$ and $s$, respectively, given by

10 $$s = \int_0^x \frac{dx}{c(x)} - t , \qquad l = x . \qquad\qquad 6$$

The transformed eKdV is then

$$\zeta_l + \frac{1}{c^2\sqrt{Q}}(\alpha + \alpha_1\zeta)\zeta\zeta_s + \frac{\beta}{c^4}\zeta_{sss} = 0 \qquad\qquad 7$$

and $\qquad\qquad \zeta = \eta\sqrt{Q(l)}. \qquad\qquad 8$

The transformation scales time so that disturbances traveling at the linear speed, $c$, remain at constant $s$.

The system is often referred to as a slowness coordinate system.  Because $\zeta$ varies relatively slowly in $l/c$





compared to $s$, terms such as $c\zeta_l$ are neglected relative to $\zeta_s$. The transformed KdV equation is the same

as the transformed eKdV equation with $\alpha_1 = 0$.

Important solutions of the KdV and eKdV equations are waves of permanent form. One family of

these waves are the solitary waves. There is a strong tendency for a long but otherwise arbitrary initial

condition to evolve into a train of solitary waves (e.g. Lee & Beardsley, 1974; Drazin & Johnson, 1989).

The solitary wave solution for the KdV equation in ($l$, $s$) space for constant parameters is given by (Zhou

& Grimshaw, 1989)

$$\eta = \eta_0 \operatorname{sech}^2\left(\dfrac{\dfrac{c-V}{c}l + cs}{\Delta}\right) \quad \text{where} \quad V = c + \dfrac{\alpha\eta_0}{3}, \quad \Delta^2 = \dfrac{12\beta}{\alpha\eta_0}. \qquad 9(a, b, c)$$

The solitary wave is a single 'bump' propagating at speed $V$ without change in form non-linearity being

balanced by dispersion (Fig.1(a)). The amplitude, $\eta_0$, is inversely proportional to the square root of the

width, $\Delta$ – higher amplitudes imply narrower widths. The solitary waves can be either waves of elevation

($\eta_0 > 0$, $\alpha > 0$) or waves of depression ($\eta_0 < 0$, $\alpha < 0$). Since the product $\alpha\eta_0$ is always greater than zero,

KdV solitary-like waves always travel with wave speed greater than $c$, Eq. (9b). For future use, it is useful

to consider the difference between the magnitude of the nonlinear and dispersive terms:

$$\chi = \left|\dfrac{\alpha}{c^2}\eta\eta_s\right| - \left|\dfrac{\beta}{c^4}\eta_{sss}\right| \qquad 10 \text{ (a)}$$

and the analytical values of these terms for $\eta = \eta_0\operatorname{sech}^2(\xi)$, where $\xi$ is the argument given in Eq. (9a),

are:



$$\frac{\alpha}{c^2}\eta\eta_s = -\frac{2\alpha}{c\Delta}\eta\eta_0 \operatorname{sech}^2(\xi) \qquad \text{10 (b)}$$

$$\frac{\beta}{c^4}\eta_{sss} = 8\frac{\beta}{c\Delta^3}\eta_0\left(\operatorname{sech}^2(\xi)\tanh(\xi)\right)\left(2\operatorname{sech}^2(\xi) - \tanh(\xi)\right). \qquad \text{10 (c)}$$

Note that for $\operatorname{sech}^2(\xi)$ the nonlinear term is for the most part larger than the dispersive (Fig. 1(b)).

The solitary wave solution to the eKdV equation has a more complicated analytical form (Stanton

& Ostrovsky, 1998):

$$\eta = -\frac{\alpha\nu}{2\alpha_1}\left\{ \tanh\left(\frac{\dfrac{c-V}{c}l + cs}{\Delta} + \sigma\right) - \tanh\left(\frac{\dfrac{c-V}{c}l + cs}{\Delta} - \sigma\right)\right\} \qquad \text{11(a)}$$

where $\nu$ is a nonlinearity parameter between zero and one, and the other parameters are

$$V = c - \frac{\alpha^2\nu^2}{6\alpha_1} \;;\; \Delta^2 = -\frac{24\alpha_1\beta}{\alpha^2\nu^2} \;;\; \sigma(\nu) = \frac{1}{4}\ln\left[\frac{1+\nu}{1-\nu}\right]. \qquad \text{11(b,c,d)}$$

The shape of the 'tanh' eKdV solitary wave is similar to the 'sech$^2$' KdV solitary waves for small

amplitude (Fig.1c). As amplitude increases the eKdV solitary waves become thicker than the KdV

solutions. Unlike the sech$^2$ solitary wave, the tanh wave has a maximum amplitude, which is given by

$\alpha/\alpha_1$ (e.g. Stanton & Ostrovsky, 1998). For our application we assume sinusoidal tidal forcing at the

boundary $l = 0$, Eq. (12a), where $a_0$ is tidal amplitude and $\omega$ is the frequency of the internal tide, with

periodic conditions Eq. (12b)

$$\zeta(s, l = 0) = a_0 \sin(\omega s) \qquad \zeta\left(s + \frac{2\pi}{\omega}, l\right) = \zeta(s, l). \qquad \text{12(a,b)}$$



We employed the same finite difference scheme as Holloway et al. (1997) to solve the eKdV Eq.

(7) numerically. The finite difference scheme is a central difference method, (e.g. Lapidus & Pinder,

1982), which was first developed for the KdV equation by Berezin (1987), and for the variable

coefficients KdV by Pelinovsky et al. (1977). The difference scheme for the generalized KdV equation

5 remains numerically stable provided

$$\frac{\Delta s}{\Delta l}\left(|\alpha\zeta| + \frac{3\sqrt{3}\beta}{2(\Delta l)^2}\right) < 1 \qquad\qquad 13$$

where $\Delta l$ and $\Delta s$ are grid resolution spacing values in space and time, respectively, (e.g. Holloway et al.,

1997). Note values of $\Delta s$ = 55s and $\Delta l$ = 10 m are used throughout this work.

## 3. Two-Layer Model

We are interested in modeling the evolution of the internal tide as it propagates shoreward from the

shelf break. Since the greatest oceanic signal is the first internal mode, the stratification of the

continental shelf/slope region is modeled as a two-layer fluid. This approximation greatly simplifies the

problem; the numerical scheme is much less complex for the two-layer case than the continuously

15 stratified case, and the results are easier to interpret. Using the two-layer model, we study the

propagation of the internal tide over various types of topography, including the simplest case of flat

bottom with level interface and progressing to realistic topography with sloping interface. All cases

have been run within the quadratic nonlinear framework of the KdV equation, and the results are

compared with the eKdV model.





For the KdV and eKdV Eqs. (1, 2) to be valid, the leading two terms must constitute the primary balance. The nonlinear and dispersive terms can become important, but the assumptions leading to the KdV and eKdV equations are violated if either of the nonlinearity or dispersion terms approach the magnitude of the leading terms. Nonlinear transformation of the internal tide leads to the generation of nonlinear waves which tend to become solitary-like in form as the dispersive term becomes important.

We begin by discussing the coefficients of the KdV and eKdV equations for a two-layer fluid, where the density difference between the layers is chosen to be a constant: $g\Delta\rho/\rho = .014$ m/s$^2$, a representative value for the Coastal Mixing and Optics (CMO) experiment (Levine & Boyd, 1999), for example at a mooring in the Middle Atlantic Bight located at 40.5ºN, 70.5ºW, and also in agreement with the stratification near the mooring location displayed in Barth et al. (1998). The linear phase speed, $c$, is then a function of $h_1$ and $h_2$ only, Eq. (3a) and Fig. 2a, with values of $c$ symmetric about the line $h_1 = h_2$, since the parameter for the effective or harmonic depth,

$$h' = \frac{h_1 h_2}{h_1 + h_2} \qquad\qquad 14$$

is contained within the phase speed (Apel, 1987) and lines of constant total water depth are perpendicular to the line $h_1=h_2$. For a given total water depth, the speed is greatest when $h_1=h_2$ and decreases as difference in layer thickness increases. Starting at a point on the line $h_1=h_2$ and keeping the thickness of one of the layers constant, the speed of the wave decreases as the thickness of the other layer decreases.

The coefficient of the non-linearity term, $\alpha$, is also a function of $h_1$ and $h_2$ only, Eq. 3b and Fig. 2b. The values of $\alpha/c$ are anti-symmetric about the line $h_1=h_2$, where $\alpha/c=0$. Starting at a point on the line $h_1=h_2$ and keeping the thickness of one layer constant, the value of $\alpha\rightarrow\infty$ as the thickness of the other

layer decreases. The absolute value of $\alpha/c$ changes least rapidly when $h_1 \approx h_2$. When the thicker layer is larger than the thinner layer by at least a factor of 2-3, then $\alpha/c$ is relatively insensitive to the thickness of the thick layer, that is when $h_2 >> h_1$, then $|\alpha/c| \approx 3/2h_1$ and is not a function of $h_2$. $\alpha/c$ is also important since when multiplied by the amplitude, $\eta$, it represents the ratio of the nonlinear to the linear terms in

the KdV Eq. (1).

The coefficient of the dispersive term, $\beta$, divided by $c$ is also a function of $h_1$ and $h_2$ only, Eq. 3c and Fig. 2c, whose values are symmetric about the line $h_1=h_2$. The value of $\beta/c$ for any given water depth is a maximum when $h_1=h_2$, values decrease as either of the layers becomes small.  The interpretation of Fig.2c as a ratio of terms is complicated since, unlike Fig.2b, the derivatives do not cancel and the ratio

cannot be simplified.

The coefficient of the cubic nonlinear term, $\alpha_1$, when divided by $c$ is also a function of $h_1$ and $h_2$ only, Eq. (3d) and Fig. 2d. $\alpha_1$ is always negative and is symmetric around the line $h_1=h_2$, while for a given water depth the magnitude of $\alpha_1$ is least when $h_1=h_2$.  The value of $\alpha_1 \rightarrow -\infty$ as either one of $h_1$ or $h_2 \rightarrow 0$. It is also useful to calculate the ratio $\alpha/\alpha_1$, see O'Driscoll (1999). The relative importance of the quadratic

to cubic nonlinearity is given by $\alpha/\alpha_1 \eta$. For a given water depth cubic nonlinearity is most important when $h_1 \approx h_2$, i.e. when the magnitude of $\alpha$ is small.  The magnitude of the quadratic nonlinear term is much greater than that of the cubic nonlinear term when the water depth of one layer is much greater than the other and in this case the eKdV model is very similar to the KdV model.



### *3.1* **The Korteweg de Vries (KdV) Model solutions**

Using the KdV equation, we first investigate 4 cases with level bottom for different combinations of $h_1$ and $h_2$. We then progress to constant sloping bottom, with both horizontal and sloping interface. Finally, we make model runs with realistic topography at the sites of the CMO.

### 3.1.1 Level Bottom

We begin by studying the evolution of the internal tide over a level bottom, with level interface ($h_1$ and $h_2$ constant). This simple fluid arrangement is instructive when developing an intuitive feel for the generation and propagation of internal wave packets. A level bottom is also a good approximation for the

10 continental shelf where the total water depth changes slowly in the horizontal. Four cases (Cases 1-4) using different layer thickness were selected to look at the effects of different relative magnitudes of $\alpha$ and $\beta$ (Table 1). Case 1 ($h_1$=50m, $h_2$=150m) was chosen because these are reasonable values of upper and lower layer depth on outer continental shelves. Results are shown in Fig. 3a-b and Fig. 4a-c. A sinusoidal internal tidal amplitude of 5 m was used as the forcing at $l = 0$, Eq. 12. As it propagates, the internal tide

steepens on the trailing edge or "back face" of the sine wave and a shock-like front forms at about $l \approx 50$ km. At this stage we also see the beginnings of undulations developing behind the shock-like front. By 75 km a pack of nonlinear solitary-like waves has begun to form. It is evident at 100 km that the solitary-like waves of depression are rank ordered with the largest amplitude first. The first three waves at 100 km are compared with the sech$^2$ solitary wave-form in Fig. 4a. Using the values of $\alpha$, $\beta$ and $c$, the shape

of the solitary wave, Eq. (10), is determined by a single parameter, the amplitude, which is subjectively adjusted for best fit. The fact that the subjectively chosen sech$^2$ fits contain an arbitrary offset does not

prevent the waves from being exact solutions to the KdV equation provided they propagate locally and at the local offset depth, as was shown by Zabusky & Kruskal (1965). The smaller amplitude trailing waves, Fig. 4b, appear more symmetric and sinusoidal in shape compared to solitary waves. Fig. 3b is a contour plot of displacement showing propagation of the internal tide in ($l,s$) space. The transformation to $s$ and $l$

space results in a wave of speed $c$ following a line of constant $s$. A solitary wave with phase speed $V > c$ will appear at smaller value of $s$ for increasing $l$ along the propagation path, i.e. will curve to the left as the tide progresses vertically up the plot along a maximum /minimum. The maxima and minima of the nonlinear waves travel at different speeds with the leading one (at smallest $s$) traveling fastest. Since a given solitary-like wave may vary in amplitude as it propagates in $l$, we also expect the track of the wave

to curve in $s$ and $l$ space. The trailing sinusoidal-like waves travel with wave speed less than $c$, indicating that dispersion is important. An interesting observation is that some of the minima of the nonlinear waves initially travel with wave speed less than $c$ and eventually travel with speeds greater than $c$ (for example see 7th min. in Fig.3b).

Fig. 4c shows the difference between the magnitudes of the nonlinear and dispersive terms, $\chi$

(Fig.1b), for Case 1 at various distances in $l$. At 100km we see the balance changing over the tidal period. The leading waves at the left have the shape expected for a $sech^2$ solitary wave (Fig. 4a). The trailing waves to the right appear more sinusoidal in shape, and are relatively more dispersive than a $sech^2$ wave. Upwards of twenty waves have formed when the internal tide has traveled 160 km. The leading six to seven waves travel with speed greater than $c$ and have a nearly $sech^2$ form. The trailing waves travel

slower than $c$ as expected for waves that are more dispersive.





For Case 2 we choose $h_1$ and $h_2$ such that $\alpha/c = -.02$ as in Case 1, but the value of $\beta/c$ is less than

half that of Case 1. Since the ratio of the dispersive coefficient to the nonlinear coefficient has been

reduced by more than half, we expect Case 2 to be more nonlinear, the internal tide to steepen sooner,

and nonlinear internal waves to form at smaller $l$.  Fig. 5 shows that the internal tide evolves similarly to

Case 1.  However, as expected, the shock-like front and subsequent undulations appear sooner (smaller

values of $l$). Comparing Figs. 3a and 5a, the internal tide is more nonlinear at $l = 50$ km for Case 2 than it

is for Case 1, while a greater number of solitary waves have formed in Case 2 by $l=100$km, and are more

closely spaced. More of the leading waves have $\mathrm{sech}^2$ form at 165 km in Case 2 when compared to Case

1.  For Case 2, a few more of the leading waves travel with speed greater than $c$ compared with Case 1;

the remaining waves have speed less than $c$, and disperse from the leading waves as $l$ increases.

For Case 3 we choose $\beta/c = 1250$, as in Case 1, but with $\alpha/c = -.0021$, a factor of ten less than the

value used in Case 1 and 2. As a result, we expect the internal tide to be much less

nonlinear. Indeed, the internal tide steepens slowly and even by $l = 200$ km solitary-type waves have not

been generated (see O'Driscoll 1999).

For Case 4 the nonlinearity parameter is half that used in Case 1, $\alpha/c = -.01$ and $\beta/c$ is the same

value. So, we expect the resultant internal tide to be more nonlinear than Case 3 but less so than either of

Cases 1 or 2. The internal tide steepens slowly and the first wave of depression begins to form when $l \approx$

100 km (Fig. 6). Fewer nonlinear waves have formed at this point than in either Case 1 or Case 2.  By $l$

= 200 km only two or three solitary-like waves have formed.



### 3.1.2 Constant Bottom Slope

The propagation of the internal tide along constant sloping topography was studied for cases of constant upper layer thickness (Case A) and sloping interface (Case B), both of which are possible on continental shelves. We have chosen starting layer thickness at $l=0$ the same as Case 1 for a flat bottom,

i.e. $h_1=50$m and $h_2=150$m, with bottom slope of 1/1000 so that total depth decreases from 200 to 0 m over a distance of 200 km.

We first investigate the case of constant sloping bottom with constant upper layer thickness (Case A). The value of $c$ decreases in shallow water, while $\alpha \rightarrow 0$ as $h_2 \rightarrow h_1$ at water depth of 100 m (Figs. 2, 7a). Seaward of this depth, where $h_2 > h_1$, $\alpha < 0$ and solitary waves are waves of depression, whereas

shoreward of this depth ($h_2 < h_1$), $\alpha > 0$ and solitary waves exist as waves of elevation only. $\beta \rightarrow 0$ as the product $h_1 h_2 \rightarrow 0$. Since the magnitude of $\alpha$ is initially relatively large we expect the sinusoidal internal tide to transform rapidly resulting in the formation of several nonlinear waves (as previously seen for the flat bottom cases). Since $\alpha \rightarrow 0$, these waves may not be so nonlinear as to violate the weakly nonlinear constraint on the KdV model. However, since the value of $\alpha$ rapidly increases for $l>100$km, we expect

the waves of elevation to become highly nonlinear thereby possibly violating the weakly nonlinear condition.

Figs. 7b and c show the internal tide signal for Case A at different values of $l$, i.e. at different water depths. The internal tide steepens and rapidly becomes nonlinear, resulting in the generation of a shock-like front and subsequent undulations by $l \approx 50$km. Shoaling further, the internal tide becomes more

nonlinear with the oscillations starting to resemble solitary waves by $l=70$km. However, unlike Cases 1, 2, and 4, the waves never develop into mature internal solitary waves as the magnitude of $\alpha$ continually



decreases. By $l = 90$ km the waves resemble a symmetric, dispersive packet, as further evidenced by Fig.

8a. Initially the relatively large magnitude of $\alpha$ resulted in the rapid steepening of the internal tide, so

much so that the Case A tidal signal at $l$=50km resembles those of both Case 1 and Case 2 for flat bottom.

However, as the value of $\alpha \to 0$ the nonlinear waves are prevented from developing into solitary waves,

since higher order terms (neglected in KdV) become of order $\alpha$ or larger and thus cannot be ignored,

thereby rendering the KdV model invalid in this neighborhood.  At $l$=100km the packet certainly looks

symmetrical about a horizontal axis, that is to say the waves are neither polarized as waves of depression

nor elevation, since KdV solitary waves cannot exist when $\alpha=0$. At 115km the waves have switched

polarity; they have become waves of elevation, a result of $\alpha$ having become positive. This transition is

seen in Fig.8b where the leading waves are compared with sech$^2$ solitary form. Beyond 100km the waves

rapidly approach solitary waves of elevation since $\alpha$ becomes large quickly.

As the internal tide propagates into shallow water the leading face of the wave steepens but, unlike

cases 1-4, the decreasing magnitude of $\alpha$ causes this steepening to slow down and there is virtually no

change in wave slope steepness between 70 and 90 km.  The rate of change of the slope of the leading

face changes sign when $\alpha$ becomes positive and the slope steepens rapidly, while the back face of the

internal tide slackens. The steepening of the leading wave will lead to the formation of a second shock-

like front (or a "reverse hydraulic jump" as has been described by Holloway et al., 1997). Fig. 7c gives a

clear picture of the wave speed.  The leading solitary-type wave initially travels with speed very slightly

greater than $c$ but becomes slower than $c$ when $l \sim 90$km.  The second solitary-type wave also has initial

speed greater than $c$ but becomes slower than $c$ when $l \sim 80$km. All of the other waves travel with phase

speed less than $c$. For values of $l > 100$km all waves travel with speed less than $c$, the reason becomes



clear upon examination of Fig. 8a which plots the difference between the magnitudes of the nonlinear and dispersive terms for Case A.

The leading waves are slightly more nonlinear than dispersive when $l \approx 70$ km but become less so as $l$ approaches 100 km. When $\alpha=0$ ($l=100$km) the value of the nonlinear term is zero and the waves look like a dispersive packet. Since $\alpha > 0$ for $l > 100$ km, the nonlinear term is again a factor and the waves become a hybrid by $l=115$km, interchanging back and forth across the length of the wave between being more nonlinear and dispersive. The waves travel slower than $c$ since the magnitude of the dispersive term is slightly greater than the nonlinear term.

For Case B with constant sloping bottom and sloping upper layer, we also begin in 200m water with $h_1 =50$m and $h_2 =150$m. For this case the bottom slope is again 1/1000 and the interface slope is 1/4000 with the result that both layers vanish simultaneously at 200 km. The values of the KdV parameters are shown in Fig. 9a. The magnitude of $\alpha$ increases from $l=0$ all the way to the shallowest water, unlike Case A where $\alpha$ passes through zero, so we expect the internal tide to become nonlinear sooner than for Case A, and any solitary waves to remain as waves of depression. We do, however, expect the waves to become unstable, a result of the increasing magnitude of the nonlinear parameter combined with the decreasing value of the dispersive parameter. This combination of events will result in the weakly nonlinear, dispersive KdV becoming invalid at $l = 95$ km. Fig. 9b is a plot of the internal tide for Case B at several values of $l$. The internal tide steepens rapidly and a shock-like wave, followed by undulations, has evolved from the transforming tide by $l=40$km. The internal tide continues to steepen and several nonlinear waves have formed by $l=55$km. These leading nonlinear waves mature into rank ordered solitary waves by 65km. Fig. 9c shows that most of the solitary waves eventually travel at a phase speed


greater than $c$. The waves are more nonlinear than dispersive and the increasing value of the nonlinear

parameter combined with the diminishing value of the dispersive parameter leads to the model becoming

numerically unstable (O'Driscoll 1999).

### 3.1.3   Realistic topography and stratification

We now proceed to the transformation of the internal tide for the case of realistic topography for

the CMO site.

The CMO site was located in the Middle Atlantic Bight. CTD profiles were made across the continental

shelf from shallow water to beyond the continental slope. Boyd et al. (1997) have concluded that the

internal tide at the site is primarily a first mode internal wave, further justifying our choice of a two-layer

model. An upper layer thickness of 25 m is a representative average value for the duration of the

experiment (July and August 1996). The line 'CMO' in Fig.2 shows the values that the KdV parameters

take as the internal tide propagates across the continental shelf. Since the upper layer, $h_1$, is constant, the

'CMO' line will be straight, but since the total depth does not vary linearly in $l$, the value of $h_2$ does not

change linearly along this line.

Fig. 10a shows the values of the KdV parameters as a function of $l$. Though undulating, the bottom

topography is similar to the constant sloping bottom cases. Recall that we chose an upper layer depth of

50 m for Case A, whereas here we have chosen $h_1 = 25$ m. $\alpha$ starts out negative with relatively large

magnitude. The magnitude decreases, similar to Case A, changing sign as the bottom shoals and $h_1 > h_2$

when the value increases rapidly. Values of $\beta$, $c$ and the horizontal variability parameter, $Q$, are similar

to Case A. Figs. 10b and c show results for tidal forcing of amplitude 2 m at 180 m water depth. The



internal tide evolves similarly to Case A.  A shock-like front has formed on the back-face of the internal tide at $l$=40km.  Several nonlinear waves have formed by $l$=60km (mooring location) with the leading 4-5 waves appearing like solitary waves of depression and the trailing waves looking more like a dispersive packet.  Several more waves have formed by 80km but the number of solitary-like waves seems to have been reduced to the leading two waves.  All of the trailing waves do appear as a dispersive packet since the magnitude of $\alpha$ has decreased.  More waves continue to form but by 100 km the packet is neither a pack of waves of elevation nor depression, not unlike Case A.  Beyond $l$=125km, large $\alpha$, the waves reverse polarity and rapidly develop into mature solitary waves of elevation.  The results show that the CMO case and Case A are similar, though more solitary waves have formed for the CMO case, due to the fact that at the CMO site the value of $\alpha$ is initially twice that of Case A.  The internal tide becomes unstable beyond $l$=130km, a result of the increasing value of the nonlinearity parameter combined with the vanishing dispersion parameter. Fig. 10c is a plot of the evolution of the internal tide as it propagates over the continental shelf, increasing in $l$. The leading solitary-like waves initially travel with speed very slightly greater than $c$, as in Case A.  The waves slow down to travel at speed $c$ where $l \approx 90$km and $\alpha$ is very small.  The speed of the waves then becomes slightly slower than $c$ but faster and more complicated than Case A, due to the undulating topography.

The difference in magnitudes of the nonlinear and dispersive terms, $\chi$, is plotted in Fig. 11.  The leading 2-3 three waves are initially more nonlinear than dispersive but the diminishing magnitude of $\alpha$ leads to the waves becoming more dispersive-like and the waves begin to slow down. The negligible value of $\alpha$ between $l$=100-115km results in the waves behaving very much like a dispersive packet and they travel with wave speed slightly less than $c$.  The increasing value of $\alpha$ after it passes through zero,

leads to the nonlinear term becoming almost the same order of magnitude as the dispersive term before

the model becomes numerically unstable shortly beyond $l$=130km.

## 3.2 The extended Korteweg - de Vries (eKdV) model

All of the model runs discussed in section 3.1 were also made using the extended Korteweg-de

Vries (eKdV) equation. The ratio of the nonlinear parameters, $\alpha/\alpha_1$, is the theoretical maximum amplitude

for the solitary wave solution to the eKdV solution. The ratio of the quadratic to cubic nonlinear terms in

the eKdV equation depends upon the displacement height, $\eta$ and is given by $\alpha/(\alpha_1\eta)$. For flat bottom,

Cases 1, 2 and 4, the maximum amplitude $\eta_0$ ~18m for the KdV numerical solution. For Case 1, $\alpha/\alpha_1 >$

40 and therefore the nonlinearities result predominantly from the quadratic nonlinear term, see O'Driscoll

(1999). For Cases 2 and 4 the magnitude of $\alpha/\alpha_1$ is just over 20 m, and both the quadratic and cubic

nonlinear terms will be important.

For the case of sloping bottom with horizontal interface, Case A, the ratio $\alpha/\alpha_1$ passesthrough

zero ($h_1$=$h_2$) and we expect the cubic nonlinear term to be important. The results of this model run are

shown in Fig. 12. The internal tide evolves similar to the KdV case (Fig. 7b) with a shock-type wave

followed by several nonlinear oscillations on the back face of the internal tide at $l$=50km. The internal

tide in both frameworks look similar at 70 km where several nonlinear waves of depression having

formed. The KdV solitary-like waves flip polarity at 100 km due solely to the fact that $\alpha$ changes sign

there.

For the CMO case, comparison of KdV and eKdV models shows a more significant difference

than for Case A. Fig. 13a-c shows the KdV and eKdV model results for a 4m internal tide having





propagated 60 km to a water depth of 69 m. The leading KdV model solitary wave (solid line) arrives at

the CMO central mooring ~0.1 tidal period ahead of the leading eKdV model solitary wave (broken line).

The KdV and eKdV models are so different at the CMO site when compared to Case A because the

magnitude of $\alpha_1$ is greater at the CMO site. Though the magnitude of $\alpha$ is less in Case A, the fact that

the magnitude of $\alpha_1$ is so small when compared to $\alpha$ means the addition of the cubic nonlinear term does

little to change the KdV results. This is not true at the CMO site where the greater magnitude of $\alpha_1$ is the

reason for the difference between the KdV and eKdV frameworks, particularly as the internal tide

propagates into shallower water and the magnitude of the ratio $\alpha/\alpha_1$ is much greater for Case A.

Comparing the leading waves from the eKdV and KdV solutions reveals a fundamental difference in

wave form; the KdV waves are taller and thinner (Fig. 13c). Solitary type solutions to the KdV ($\text{sech}^2$)

and to the eKdV (tanh) are fitted to the leading waves (Fig. 13d-e). The leading wave in the KdV model

is very well approximated by a $\text{sech}^2$ wave. The lead wave in the eKdV model is neither well

approximated by $\text{sech}^2$ or tanh, but appears to be a hybrid between the two. Fits of $\text{sech}^2$ and tanh waves

were made by subjectively choosing values of $\eta_0$ and $\upsilon$, respectively, while using the value of KdV and

eKdV parameters for 69m water depth. Note that the amplitude of the tanh wave is limited to $\alpha/\alpha_1$.

Increasing $\upsilon$ only serves to make the waves wider once the value of $\upsilon$ is close to one (Fig. 1c). The

amplitude and width of the leading waves of the packet are also compared in Fig. 14a. The width is

defined as the time it takes the wave to pass a fixed point, as measured at 42% of the amplitude. Results

from a range of different tidal amplitudes are also shown. For reference the dotted lines represent $\text{sech}^2$

and tanh for the local values of parameters $h_1$, $h_2$, and $g\Delta\rho/\rho$. For KdV the leading wave of the 2m tide

always has amplitude greater than the second and the amplitudes of subsequent waves decrease in a rank



ordered fashion. The leading wave is slightly thicker than the trailing ones which are all approximately

equal in width. For the eKdV the leading wave has larger amplitude and is thicker than the trailing waves.

For the KdV model with 4m amplitude tide all the waves fall on the same spot on the sech$^2$ curve. For

the eKdV model with 4m amplitude tide, the waves appear on the 'thick' side of the sech$^2$ curve with the

lead wave the most removed from the KdV theoretical curve. The same is true for amplitudes of 5m and

6m. The eKdV model waves appear to be evolving toward the theoretical eKdV 'tanh' curve. Note that

the amplitude of many of these waves exceeds maximum amplitude *tanh* wave of 9 m as determined by

the local parameters at the CMO site.

To learn more about the evolution of a sine wave to waves with sech$^2$ and tanh form, we ran the

model with constant parameters (flat bottom) using values at the mooring site. The runs were made with

initial tidal amplitudes of 1, 2 and 4 m in both KdV and eKdV frameworks and the width vs. amplitude

for the first and second wave in each packet is plotted at various increments of *l* (Fig. 14b). The KdV

waves grow in amplitude with approximately constant width before turning to hug the theoretical KdV

line. They then decrease in amplitude while increasing slightly in thickness. Though the KdV model

waves continue to evolve, most of them can be well approximated as being 'sech$^2$' waves after ~100km

(as was previously shown for Case 1 and Case 4). For the eKdV case, the waves are initially close to the

theoretical sech$^2$ KdV curve. The waves move slowly towards the theoretical eKdV tanh curve, ultimately

decreasing in amplitude and increasing in thickness. The last points have been plotted after the internal

tide has propagated ~240km. It appears that these waves are evolving toward tanh form, but mature over

a long distance. Also, the amplitudes of the waves are greater than the theoretical eKdV maximum but

their magnitudes decrease as the tide evolves.

Another investigation to explore the evolution in the eKdV model (constant parameters) was made using an initial condition of a sech$^2$ wave, the solitary wave solution to the KdV equation. Sech$^2$ amplitudes of 4m, 7m, 9m, and 13m (Fig. 14c) were chosen. The sech$^2$ waves are rapidly transformed to tanh waves, e.g. the 4 examples plotted reach the theoretical eKdV curve after the wave has propagated

about 10 km. A solitary sech$^2$ wave evolves much more rapidly to the tanh form (Fig. 14c), as opposed to when it is part of a packet of waves (Fig. 14b). The reason for this has not been thoroughly investigated, but provides caution for treating a packet as a group of non-interacting waves.

## 4.   Observations of Nonlinear Internal Waves

The data to be presented and discussed was collected during the CMO, for location see Fig. 15. The CMO experimental field program was conducted to increase our understanding of the role of vertical mixing processes in determining the mid-shelf vertical structure of hydrographic and optical properties. The field program was conducted on a wide shelf so as to reduce the influences of shelf break and nearshore processes. The data we discuss was collected from the CMO Central Mooring in July and August 1996,

a time when a strong thermocline is present as a result of large-scale surface heating, Boyd et al. (1997).

### 4.1   Observations during the Coastal Mixing and Optics Experiment

The Central Mooring of the CMO experiment was located at 40º 29.50' N  70 º 30.46' W in water depth of 69 m. A total of 24 temperature recorders and 5 conductivity sensors were distributed along the

mooring. Currents were measured at 14 depths from an ADCP placed a few metres above the bottom.

Boyd et al. (1997) have calculated the first mode internal wave amplitude from the velocity time



series for the period 29 July to the 31 August 1996 (year day 210 - 245, Fig. 16a). The dominant barotropic

tidal signal in the Middle Atlantic Bight is semi-diurnal, and is strongest over the period day 241-245

during spring tide (Fig. 17). A semi-diurnal signal is apparent in the first mode record, particularly during

the spring tide period. A spectrum of the first mode amplitude (Fig. 18) shows energy peak at both low

and high tidal frequencies. Much of the high frequency energy is due to bursts or pulses of high frequency

nonlinear internal waves that occur for a short period during the semi-diurnal tidal cycle. These nonlinear

internal waves propagate shoreward across the continental shelf to the south of Martha's Vineyard. The

energy at high frequency is greater over the period day 241-245 during spring tides (Fig. 18). There is a

clear maximum in energy at 2 cpd over this period, and a significant amount of energy is also contained

at 4 cpd. The energy rapidly drops for frequencies greater than 4 cpd but there is a significant increase in

energy at ~50 cpd and at ~90 cpd. To help interpret these observations, we compare them with the two-

layer eKdV model using the CMO parameters. Since we do not know where the internal tide is generated

or its amplitude, the model was run assuming a sinusoidal internal tide at distances of 24 km, 48 km and

60 km seaward of the mooring site. Three initial amplitudes of 2 m, 4 m and 6 m were used at each

distance. Fig. 14c shows the internal tide as it appears at the CMO mooring site for these nine cases. In

all cases, the leading face of the periodic sinusoidal wave slackens (or flattens) as the internal tide

propagates shoreward. This is followed by a steepening of the back face which develops into a shock-

like front. The shock-like front is followed by oscillations which subsequently evolve into a packet of

solitary-like waves.

This same pattern can often be seen in the observed time series of the first internal mode. Fig. 19

shows several individual jumps at the CMO mooring. Fig. 19a (left panel) shows first modes which best





match the model results of Fig.16c. Some features of the observations compare well with the model. The slackened leading face of the tide is always followed by a steep - almost shock like - front followed by several highly nonlinear short period waves. Although not rank ordered, the largest amplitude wave in the observed packet is always at or near the jump. The model results show that the amplitude of the jump

is greater for larger initial condition, and decreases with distance from the point of generation. Although nonlinear waves continue to evolve, their amplitudes decrease as they propagate shoreward from their generation point, and they become 'thicker', i.e. they become more tanh like. Though the modelled waves have amplitudes less than the theoretical tanh limit for local eKdV parameters, they nonetheless fit the shape of several observed waves at the CMO site.

There are also features of the observations that are not found in the model. Fig. 19 (left panel f and g) differ in that the packet that follows the shock-like front, persists until the end of the tidal period, and the waves are spread apart from each other. Fig. 19 (left panel c) shows two packets of solitary-like waves propagating past the mooring site over a tidal period. The leading slackened face is followed by a shock-like front and a packet of solitary waves. The trailing face then slackens to assume a slope similar

to the leading face but a second shock-like front, followed by a packet of solitary waves, passes before the end of the tidal period. This could be from a second internal tide front coming from another generation site, there can be overlapping semi-circles of internal wave fronts from multiple generation sites, see for example discussion in Apel et al. (1988).

Another common observation that is not found in the model results is a 'drop' in amplitude

before the jump that occurs at the beginning of the wave packet. Fig. 19 (left panel h) shows that the first internal mode drops between day 243.5 and 243.6 but the slackening slope is restored before the





arrival of the jump and packet of solitary waves. Similar 'drops' also occur in Fig. 19 (left panel b and

e) and (middle panel a and i). Another phenomenon observed is that the slope of the leading face of the

tide changes sign before the packet in several of the examples in Fig. 19 (middle panel). In Fig. 19

(middle panel h) the low frequency slope changes sign at day 236, and the solitary waves appear as

usual ahead of the trailing, low frequency signal. The signal becomes even more complicated when both

a 'drop' and low frequency slope change are present, e.g. Fig. 19 (middle panel d). In this case, the

slope of the leading slackening low frequency signal changes sign at day 242.5 and is followed by a

packet of four solitary waves. The low frequency signal is restored before the passage of a jump

followed by a packet of five large solitary waves. The trailing face retains the slope of the low

frequency signal. Fig. 19 (right panel) shows a series of jumps which are more complex than those in

the left and center panels, though they retain the basic structure of the model results over the tidal

period.

To examine the details of the wave packets themselves, the width vs. amplitude was estimated for

each wave from all events during the period day 210-245 (Fig. 20). These waves are plotted along with

the leading two waves from six of the nine model runs shown in Fig. 16c. Also shown are the theoretical

relations for solitary waves for the eKdV and KdV equations using CMO site parameters. The observed

nonlinear waves vary greatly in amplitude and width, generally having amplitudes of between 5 and 25

metres, and widths of between 200 and 600 seconds. Larger amplitude observed waves are well

approximated by model runs with large initial amplitude, particularly the 4m model. The 6m model run

from 24km seaward of the CMO site is also a very good match for several of the observed waves. A large

fraction of observed waves with amplitude less than 15m, and particularly less than 10m, are much



'thinner' than model waves with similar amplitudes. However, it seems reasonable to say that the observed waves are a good fit to the model waves.

While some features of the observations are reproduced in the model, there are many differences. The eKdV model used here is highly idealized. There are many effects that have not been

included, including bottom and internal friction, earth's rotation and mean shear. Given these limitations, we conclude that the observations are reasonably well matched by our model.

## 5. Summary and conclusions

Observations of highly nonlinear internal waves contained in the first mode time series on the mid-

continental shelf and in current meter records in shallow water have led us to investigate the transformation of the shoaling internal tide. Observations were made in the mid-continental shelf at the site of the Coastal Mixing and Optics Experiment (CMO). An existing model based on generalized KdV and eKdV equations has been simplified for use in a two-layer ocean, which is representative of realistic stratification. The model accounts for weakly nonlinear and dispersive properties of the internal tide. Earth's rotation, internal

dissipation, bottom friction, and internal shear are not included. The internal tide was forced with a periodic sinusoidal boundary condition and allowed to propagate shoreward.

The model was first run within a KdV framework with realistic continental shelf parameters. The internal tide steepens on its back face as it propagates shoreward, which is a direct result of the much greater magnitude of the nonlinear term in comparison with the dispersive term. Nonlinear waves evolve from the

internal tide after the back face forms a shock-like front. The waves appear as a rank ordered packet with the leading waves traveling fastest, since they are the most nonlinear. The leading waves usually travel

faster than the linear wave speed, $c$, and nearly fit solitary wave form for local KdV parameters ("sech$^2$").

The trailing waves usually travel slower than $c$, tend to be thinner than the local sech$^2$ waves and are

relatively more dispersive than the leading waves.

The transformation of the internal tide is dependent upon the ratio of the nonlinear to linear terms,

$\alpha\eta/c$, in the KdV equation: for greater values of this ratio the internal tide steepens sooner and nonlinear

waves are emitted sooner. The amplitude of the jump and subsequent waves is dependent upon the initial

tidal amplitude: larger tidal amplitudes imply larger jump and nonlinear wave amplitudes. For a fixed

nonlinear parameter, $\alpha$, the internal tide becomes nonlinear sooner upon decreasing the value of the

dispersive parameter, $\beta$.

The nonlinear waves are waves of depression when the nonlinear parameter, $\alpha$, is negative, and waves

of elevation when it is positive. If a packet of waves of depression propagates into a region where $\alpha > 0$ the

minima, or troughs, of the waves of depression become maxima, or peaks, of the waves of elevation as they

flip polarity.

All of the model runs made within the KdV framework were also made within the eKdV framework

which includes a cubic nonlinearity term scaled by $\alpha_1$. The results may or may not be similar, depending

upon the ratio of the two nonlinear terms, $\frac{\alpha}{\alpha_1\eta}$. If this ratio is large (greater than one) the cubic nonlinear

term is not important and the KdV and eKdV results are similar. If the ratio is of order one or less the

eKdV may evolve differently from the KdV. For most of the simulations the model results were similar

in both frameworks. However, there are some significant differences to the waves that cross the shelf

using CMO parameters. The modeled leading waves at the CMO mooring site were much 'thicker' than

sech$^2$ waves with local KdV parameters, but they had not quite developed into solitary wave solutions of the eKdV equation ('tanh').

To better understand the evolution of waves toward tanh form in an eKdV framework, without the complications of varying parameters, model runs were made using constant eKdV parameters representative of the CMO site. Upon formation, the leading waves of the packet are similar to sech$^2$ waves. The waves become 'thicker' and tend toward the tanh form upon further propagation, but never reach the theoretical tanh curve in our limited domain. To help understand why the evolution of waves from being close to sech$^2$ waves to being close to tanh waves was so slow, the internal tide was forced with a sech$^2$ wave. The evolving sech$^2$ rapidly moves to the theoretical tanh curve for all amplitudes. We conclude that the interaction between the solitary like-waves in a packet slows them from evolving into exact solitary 'sech$^2$' or 'tanh' waves.

Model runs with varying initial amplitudes and generation regions were made to help interpret the observations made at the CMO site. Some features of the observations compare well with the model. The leading face of the internal tide steepens to form a shock like front. Nonlinear high frequency waves evolve shortly after the appearance of the jump. Although not rank ordered, the wave of maximum amplitude is always close to the jump. Some features of the observations are not found in the model. Nonlinear waves can be very widely spaced and persist over a tidal period. The amplitude of the observed waves often decreases before the arrival of the jump, while the leading face may change slope before the jump arrives.

Individual observed waves were examined and the details compared to model results. The observed nonlinear waves vary greatly in amplitude and width, generally having amplitudes of between 5 and 25 metres, and widths of between 200 and 600 seconds. Larger amplitude waves are well approximated by

waves evolving from large amplitude model waves. A large fraction of smaller amplitude observed waves, particularly less than 10 m, are thinner than model waves of similar amplitude. We conclude that the observed waves are a good match to modeled waves given the highly idealized eKdV model used, and the fact that we have neglected friction, rotation and mean shear.

## Author Contribution

Kieran O'Driscoll conducted this work out while a graduate student at the College of Oceanic & Atmospheric Sciences, Oregon State University, in partial fulfillment of the degree of Master of Science. Murray Levine was the student's advisor.

## Acknowledgements

Kieran O'Driscoll would like to thank Jack Barth for a considerable review of a previous version of this manuscript. This work was supported by funding from the Office of Naval Research and Oregon State University.

**Competing interests:** Kieran O'Driscoll declares that he has no conflict of interest.

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



# Figures

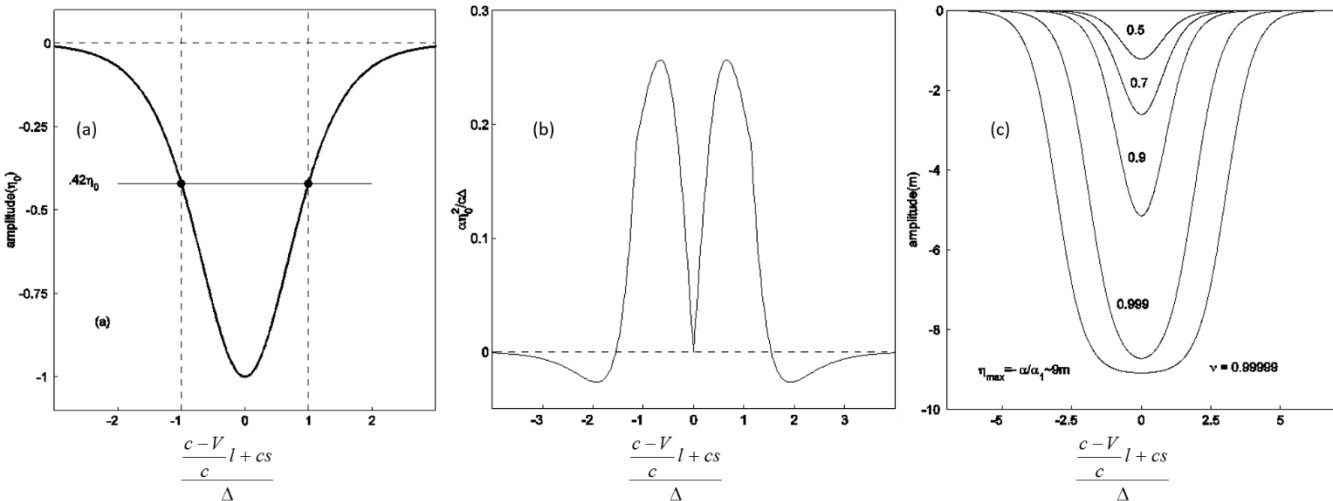

Fig1. **(a)** KdV 'sech$^2$' wave with amplitude of -1 m, where the wave argument value is shown on the abscissa and parameter values are calculated from Case 1 model runs ($h_1$ = 50 m, $h_2$ = 150 m, $g\Delta\rho/\rho$ = .014 m/s$^2$). The amplitude of these waves is reduced to 42% of its maximum value when the argument is 1. **(b)** The difference of the absolute values of the nonlinear and dispersive terms, $\chi$, in the KdV equation for the 'sech$^2$' wave shown in (a). **(c)** eKdV 'tanh' wave for several values of the nonlinear parameter, $\nu$. Parameter values as in (a).





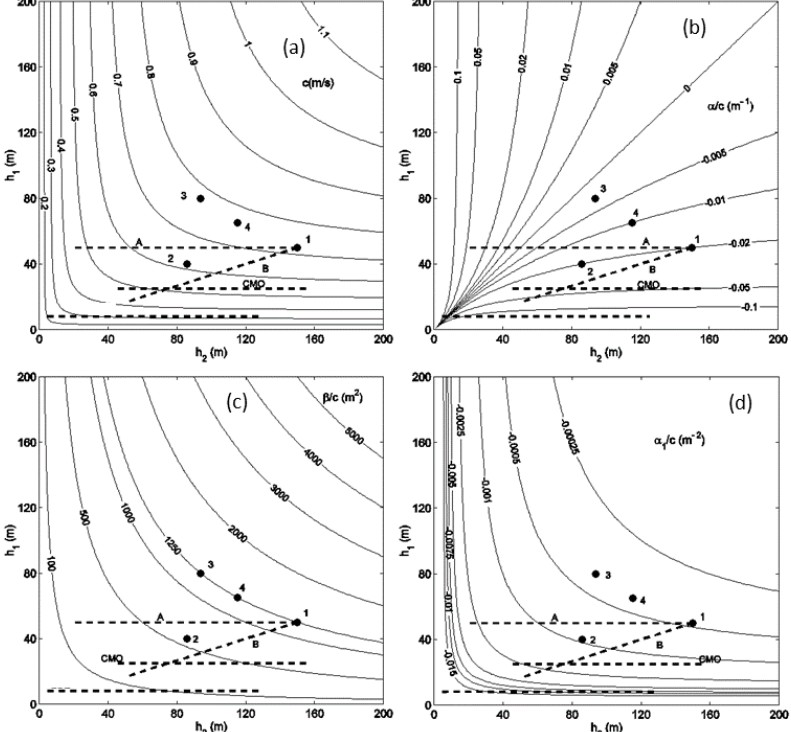

Fig. 2. **(a)** Linear wave speed, $c$(m/s), as a function of the depth of the upper layer, $h_1$, and lower layer, $h_2$. Also shown are the values for level bottom (Cases 1-4), sloping bottom (Cases A and B) and realistic slope and stratification (CMO). **(b)** KdV quadratic nonlinear parameter, $\alpha$, divided by the linear wave speed, $c$, as in (a). **(c)** KdV dispersion parameter, $\beta$, divided by the linear wave speed, $c$, as in (a). **(d)** eKdV cubic nonlinear parameter, $\alpha_1$, divided by the linear wave speed, $c$, as in (a).



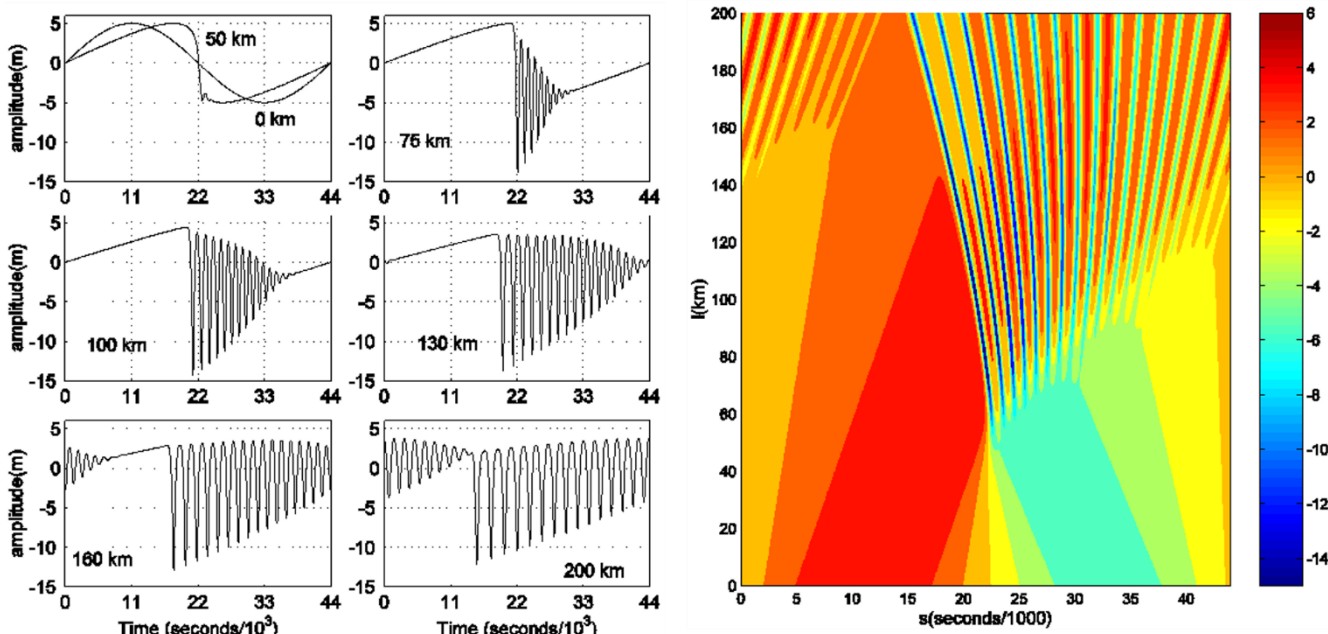

Fig. 3. Case 1 ($h_1 = 50$ m, $h_2 = 150$ m, level bottom) amplitude of the internal mode for two-layer fluid **(a)** at various distances from the boundary within KdV model framework and **(b)** as a function of distance $l$ and time $s$ within KdV model framework. The legend on the right corresponds to the amplitude of the waves (m).

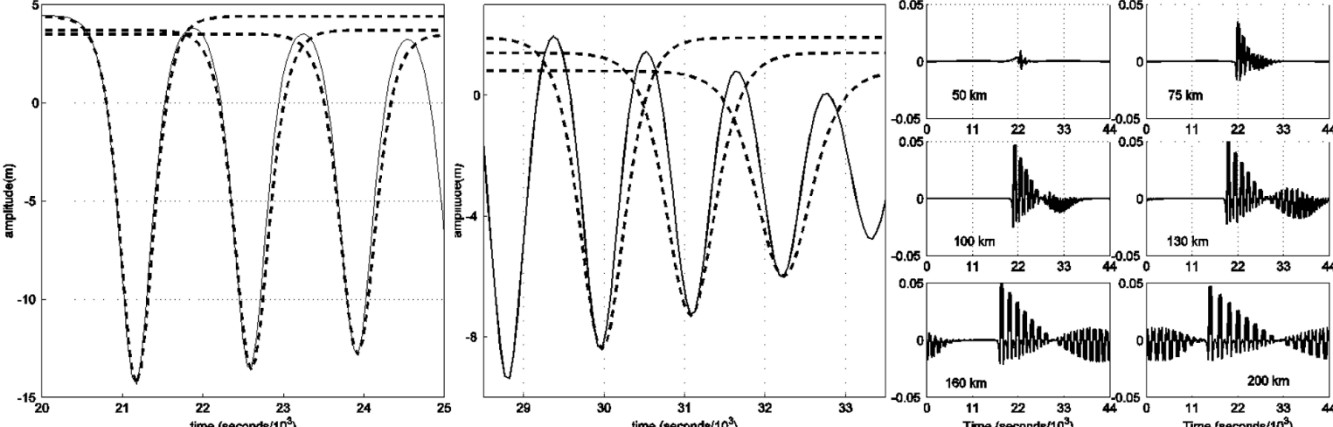

Fig. 4. Case 1. **(a)** The three leading waves of depression (solid line) at a distance of 100 km from the boundary shown in Fig. 3a are plotted with three individual sech$^2$ waves (broken lines).





**(b)** Same as (a) but for trailing waves in the packet. **(c)** Difference between the magnitudes of the nonlinear and dispersive terms, $\chi$ (non-dimensional) at various distances from the boundary within KdV model framework.

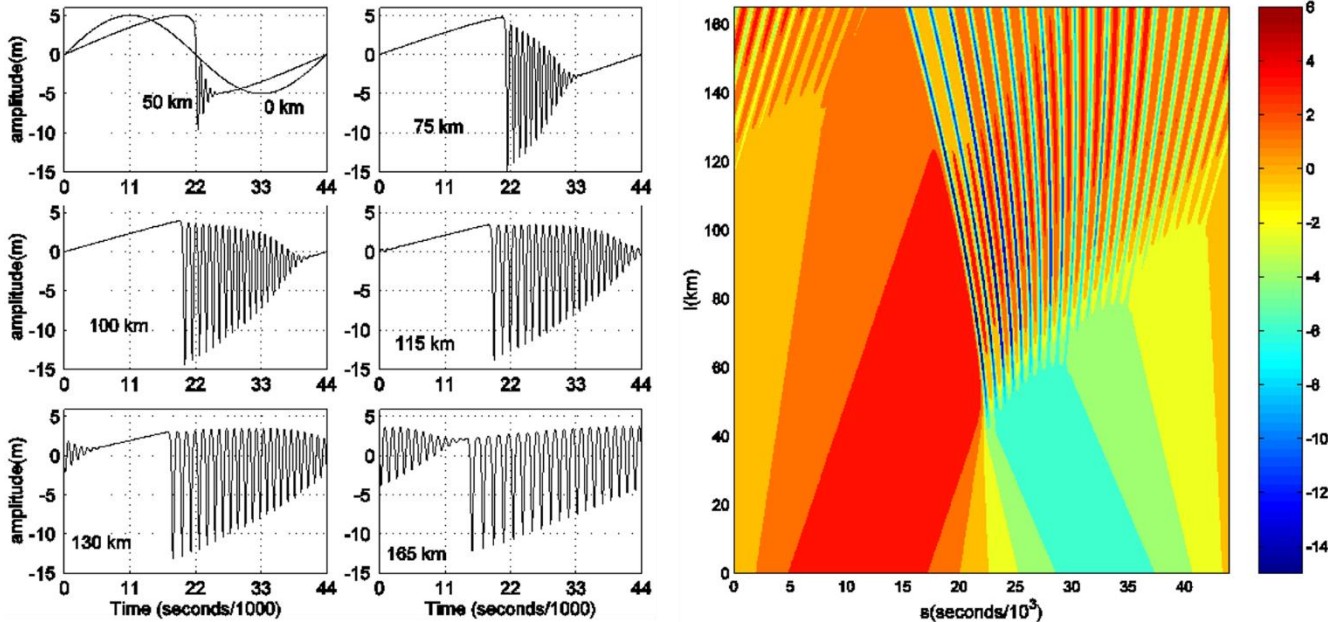

Fig. 5. Same as Fig. 3 but for Case 2 ($h_1 = 40$ m, $h_2 = 85.7$ m, level bottom).




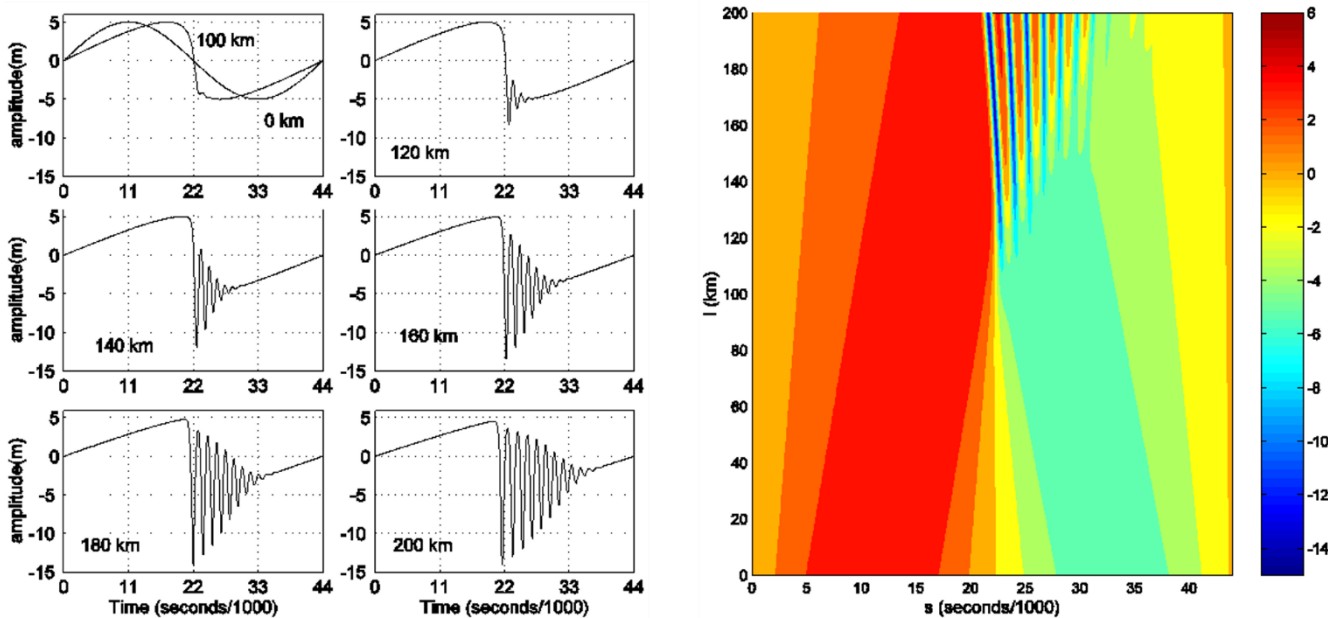

Fig. 6. Same as Fig. 3 but for Case 4 ($h_1$ = 65.1 m, $h_2$ = 115.1 m, level bottom.

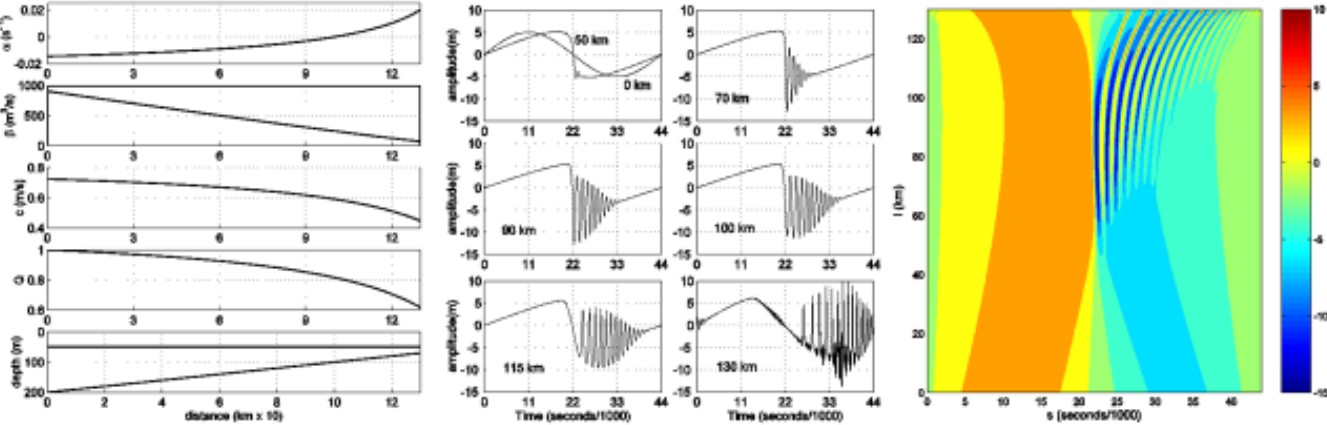

Fig. 7. Case A (constant sloping bottom with level interface, $h_1$ = 50 m). **(a, left)** KdV parameter values for quadratic nonlinear parameter, $\alpha$, dispersion parameter, $\beta$, linear phase speed, $c$, horizontal variability factor, $Q$, and depth. **(b, center)** and **(c, right)** Same as Fig. 3 but for Case A parameters.



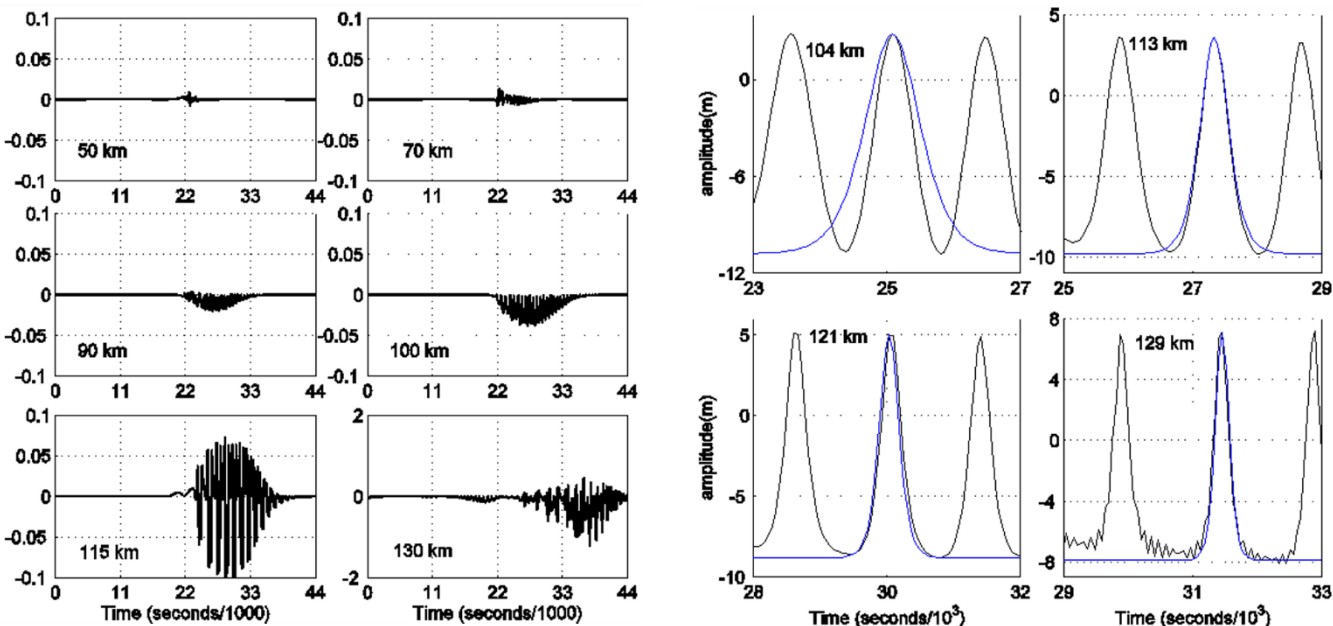

Fig. 8. Case A, **(a, left)** difference between the magnitudes of the nonlinear and dispersive terms at various distances from the boundary, and **(b, right)** leading waves of elevation (black line) at various distances greater than 100km from the boundary plotted with individual sech$^2$ waves (blue lines) within KdV model framework.

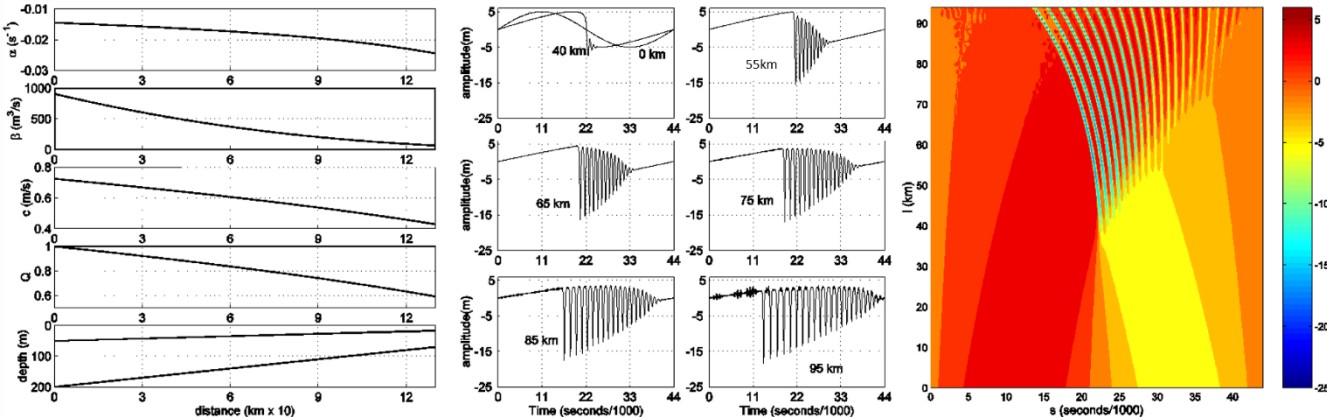

Fig. 9. Same as Fig. 7 but for Case B (constant sloping bottom with sloping interface).





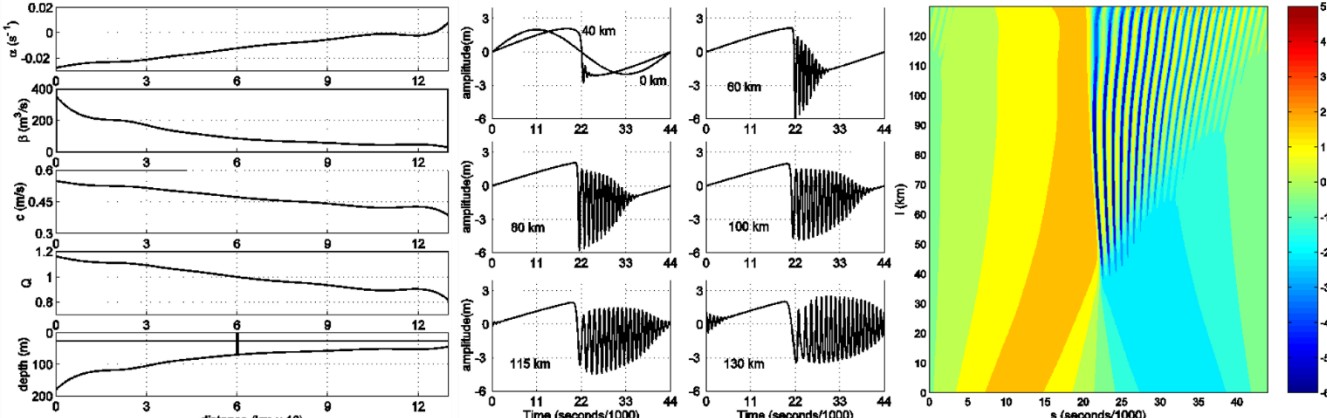

Fig. 10.  Same as Fig. 7 but for CMO experiment site (with flat interface, $h_1 = 25$ m).



Fig. 11. CMO experiment site (with flat interface, $h_1 = 25$ m) difference between the magnitudes of the nonlinear and dispersive terms at various distances from the boundary within KdV model framework.





Fig. 12. Case A (constant sloping bottom with flat interface, $h_1 = 50$ m) amplitude of the internal mode for two-layer fluid at various distances from the boundary within eKdV model framework.




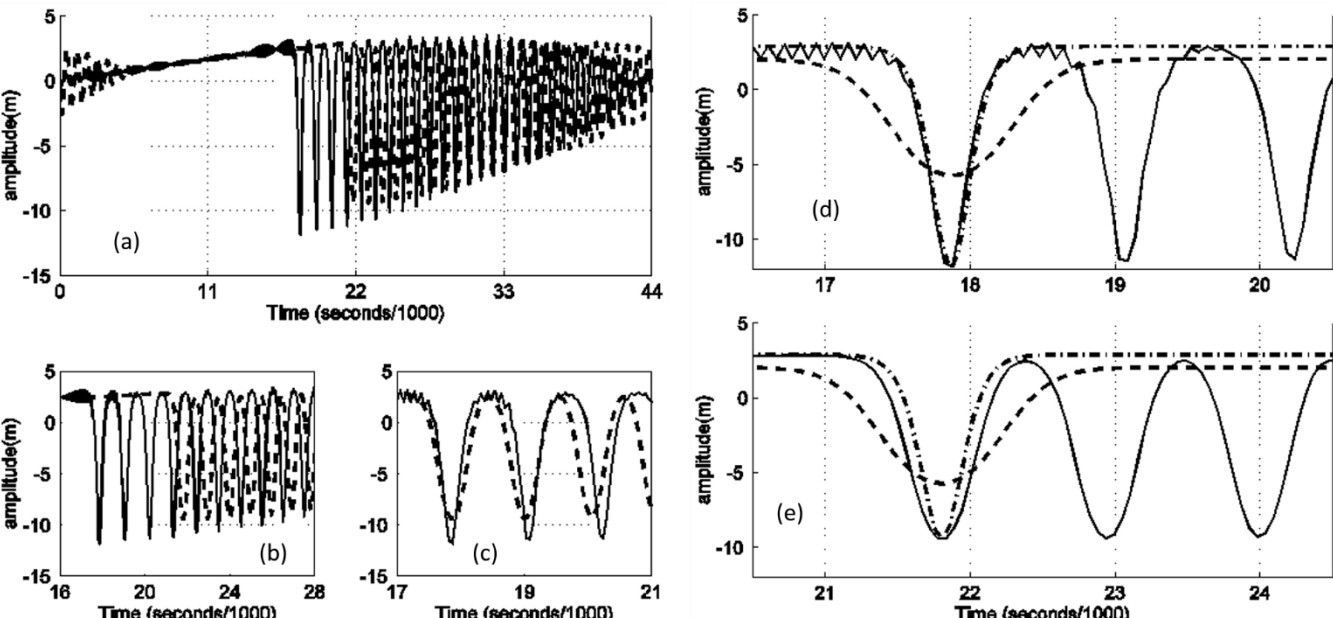

Fig. 13. CMO experiment site (realistic topography with level interface, $h_1$ = 25 m) amplitude of the internal mode for two-layer fluid at 60 km in 69 m depth water (CMO mooring site). **(a)** Comparison of KdV (solid line) and eKdV (broken line) solutions. **(b)** Close up of (a). **(c)** Leading KdV model waves (solid line) with superimposed eKdV model waves (broken line) shifted forward in time (*s*) so that the leading waves coincide. **(d)** and **(e).** The leading wave of depression (solid line) plotted with an individual sech$^2$ wave (dot-dash line) and with an individual tanh wave (dashed line) for KdV model (d), and eKdV model (e).

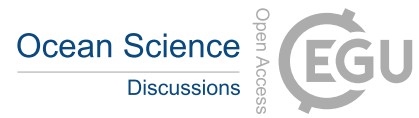

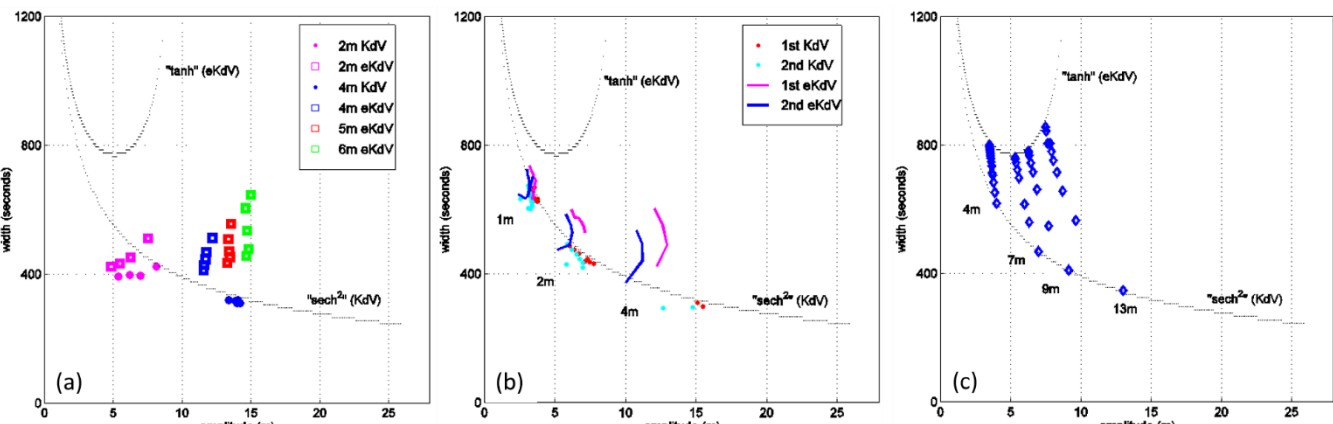

Fig. 14 **(a)** Width vs. amplitude of the leading waves of the KdV and eKdV solutions at the CMO mooring site ($h_1$ = 25 m, $h_2$ = 44m) at 60 km from the boundary in 69 m depth water. Results for initial tidal amplitudes of 2, 4, 5, and 6 m are shown. The theoretical values for $sech^2$ and tanh waves using local parameters are also shown (dotted lines). The width is calculated at 42% of the total amplitude. **(b)** Evolution of the width vs. amplitude of the two leading waves of the KdV and eKdV solutions for flat bottom ($h_1$ = 25 m, $h_2$ = 44 m) with same parameters as at the CMO site. Results for initial tidal amplitudes of 1, 2, and 4 m are shown. A value is plotted every 10 km for the 1 m tide beginning at 160 km and the lines run from 160 km to 260 km. A value is plotted every 20 km for the 2 m tide beginning at 80 km and the lines run from 80 km to 200 km. A value is plotted every 20 km for the 4 m tide beginning at 40 km and the lines run from 40 km to 100 km. The theoretical width vs. amplitude for $sech^2$ and tanh waves is also shown (dotted lines), and the width is calculated at 42% of the total amplitude. **(c)** Evolution of the width vs. amplitude of four solitary $sech^2$ waves of the eKdV solutions for flat bottom ($h_1$ = 25 m, $h_2$ = 44 m) with same parameters as at the CMO site. Results are shown for $sech^2$ amplitudes of 4, 7, 9 and 13 m. A value is plotted every 1 km up to a maximum distance of 15 km. The theoretical width vs. amplitude for $sech^2$ and tanh waves is also shown (dotted lines). The width is calculated at 42% of the total amplitude.





Fig. 15. Site of the Coastal Mixing & Optics experiment (left) located in the Middle Atlantic Bight to the south of Massachusetts. The data discussed was collected at the mooring marked 'CTD #6'.





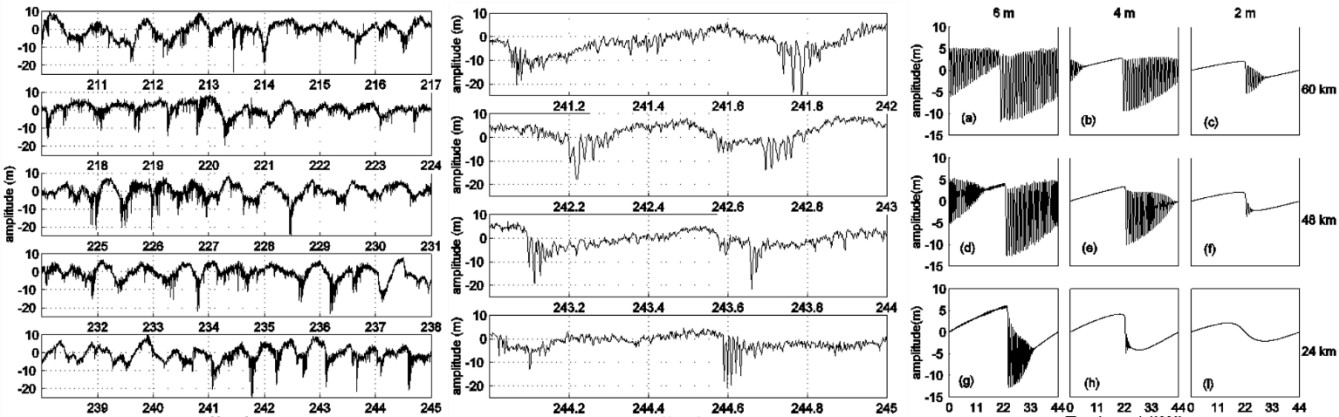

Fig. 16. **(a)** Left panel: Amplitude of the first internal mode calculated from the current meter record at the CMO mooring site over the period day 210-245 of 1996. **(b)** Center panel: Same as left panel except for the period day 241-245. **(c)** Right panel: Three sinusoids of amplitude 2 m, 4 m and 6 m, respectively, and with tidal period, as they appear at the CMO mooring site in the eKdV framework. The sinusoids have propagated shoreward from boundaries at 60 km, 48 km and 24 km offshore, respectively.







Fig. 17. Pressure (tidal) record at the CMO mooring site (top) over the period day 210 – 245 (from Boyd et al., 1997).





Fig. 18. Energy spectra of the first internal mode at the CMO mooring for the period day 210 – 245 (left), and the period 241 – 245 (right).





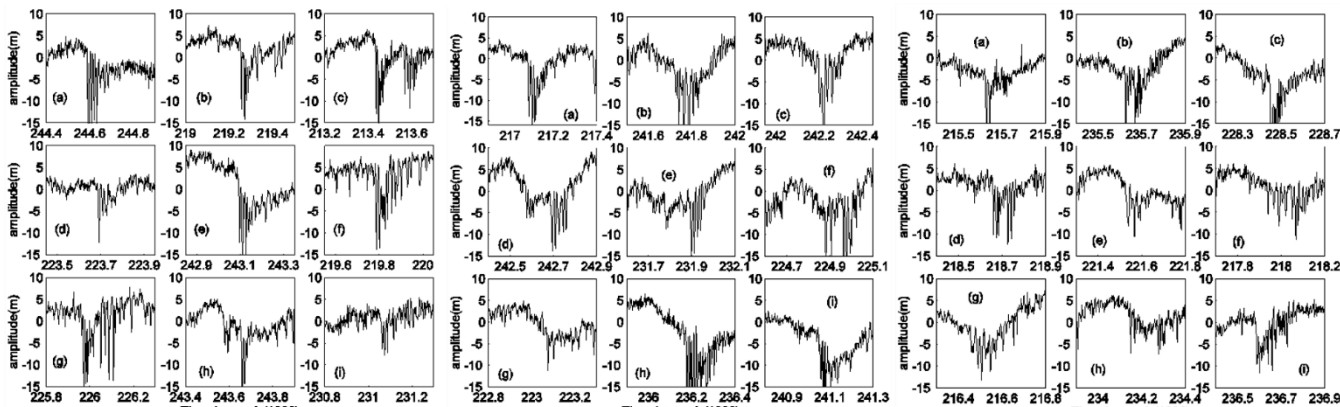

Fig. 19. Observations at the CMO Mooring site over a semi-diurnal period. **Left:** These sections of the record were chosen since they are similar to events observed over a tidal period in the model runs of Fig. 16c. **Center:** Same as Left except the record is a little bit more complicated over a tidal period.

5 Right: Same as Center.





Fig. 20. Wave amplitude vs. wave width at the CMO mooring for waves from all events during the period day 210 - 245. Also plotted are the two leading waves from six of the nine model runs shown in Fig. 16 (diamonds).



## Tables

| Case | $h_1$ | $h_2$ | $\alpha/c$ | $\beta/c$ | $\alpha$ | $\beta$ | $c$ | $(12\beta/\alpha)^{1/2}$ |
|------|-------|-------|-----------|-----------|----------|---------|-----|--------------------------|
| 1 | 50 | 150 | -.02 | *1250* | -.0145 | 906 | 0.725 | 628 |
| 2 | 40 | 85.7 | -.02 | *571* | -.0124 | 353 | 0.618 | 362 |
| 3 | 80 | 93.8 | -.0021 | *1250* | -.0016 | 972 | 0.777 | 2041 |
| 4 | 65.1 | 115.1 | -.01 | *1250* | -.0076 | 954 | 0.763 | 1227 |

Table 1. KdV parameter values for Cases 1 - 4.