# Peer review of "Simulations and observation of nonlinear internal waves on the continental shelf: Korteweg-de Vries and extended Korteweg-de Vries solutions"

_Ocean Science, 2016_

## Referee Comment (RC1) · Anonymous Referee #1 · 27 Jan 2017

Review of "**Simulations and observation of nonlinear waves on the continental shelf: KdV solutions**" by *Kieran O'Driscoll and Murray Levine*

The use of the Korteweg-de Vries (KdV) equation, and the extended Korteweg-de Vries (eKdV) equation to model the large-amplitude internal solitary waves commonly observed in the coastal ocean is now well-established, both for a constant background, and when the bottom topography and hydrology vary in the propagation direction, see for instance the recent reviews by Grimshaw *et al* (Surveys in Geophysics, vol 28, 2007, pp 273-298, Nonlinear Processes in Geophysics, vol 17, 2010, pp 663-649) and Helfrich and Melville (Annual Reviews Fluid Mechanics, vol 38, 2006, 395-425). This paper uses these models for a two-layer fluid configuration to describe the deformation of a sinusoidal internal tide into an undular bore, for a constant background case, for a linear slope, and for the topography of a transect of the Middle Atlantic Bight. The paper concludes with a comparison of model simulations with some observational data from the same region.

Much of the material presented here in section 2 and 3 is well-known and does not need to be repeated here. For the constant-coefficient KdV model the steepening of a sinusoidal wave and formation of rank-ordered solitary waves in an undular bore is now text book material, following the seminal and landmark paper of Kruskal and Zabusky in 1965. The two-layer fluid model is widely used and understood, and since the eKdV model has a negative cubic term, its solutions are known to be generally qualitatively similar to the KdV model. Likewise, the behaviour of internal solitary waves propagating over a slope has been widely studied in both the KdV and eKdV models, with often a main focus on polarity change when the coefficient of the quadratic nonlinear term changes sign. The simulations with a linear bottom slope and with the topography of the Middle Atlantic Bight have some marginal interest in that most studies have examined the behaviour of a single solitary wave, rather than a developing wave train as here, although the outcome can be understood in terms of the known behaviour of a single solitary wave, namely adiabatic deformation and transition to an elevation solitary wave train riding on a negative pedestal when the usual transition is from a negative to positive coefficient. The most interesting and novel part of the paper is section 4 where the model simulations are compared with observational data from the Middle Atlantic Bight. Although there have been several such comparisons in the literature for other sites, this would seem to be the first for this site. In summary I would recommend that the authors prepare a heavily revised and shortened paper which focusses on the material in section 4.

*Specific Comments*:

(1) Further to the comments made above, in particular most of the text in section is not needed, and neither are figures 1-6.

(2) The measure $\chi$ (10a) of the relative roles of nonlinearity and dispersion is unconventional and uninformative. A better measure is simply the ratio $\alpha/\beta$, where it should be noted that in the KdV equation (1) division by $\beta$ and a rescaling of time, clearly indicates that this is the effective measure of nonlinearity *vis-a-vis* dispersion.

(3) The transition of the steepening front into a solitary wave train is best understood using the Whitham modulation theory and asymptotic solution, as developed by Gurevich and Pitaevskii. It is well known that, at least in the KdV model, the leading waves are solitary waves. The detailed discussion on this aspect is not needed here.

(4) The large-amplitude solitary wave solutions of the eKdV equation are more usually called "table-top" waves than the term "tanh" used in the text.

(5) The title should mention "internal" and should not use an acronym.

---

## Referee Comment (RC2) · Anonymous Referee #2 · 21 Feb 2017

[11pt]article amsmath,times,epsfig,eepic,color

*Review of*
***"Simulations and observation of nonlinear waves on the continental shelf: KdV solutions"***
*by K. O'Driscoll and M. Levine*

This paper compares solutions of the KdV and eKdV (KdV plus cubic nonlinearity) using stratifications and water depths based on observations from the Middle Atlantic Bight. Two sets of simulations are done: one set uses a flat bottom and compares

solutions in different regions of parameter space. The second set investigates shoaling waves over a constant bottom slope. The eKdV model was then used to do some simulations more closely based on the observations (thinner upper layer, realistic bathymetry).

I have a number of problems with the paper and think that is requires significant revision. My basic problem is that there doesn't appear to be much that is new here other than the application to the CMO site in the Middle Atlantic Bight and that is quite a small part of the paper. The paper needs considerable polishing. Figures are in some cases hard to read and many dimensional values are given without units. It would probably benefit from being shortened and more focussed on the comparisons with observations however I am not convinced of the value of these simulations in that context. The authors make some comparisons of their results with those of Holloway et al from 30 years ago. Recent work has been done in this area using model equations that include rotation (e.g., Grimshaw and co-workers). The authors need to make a compelling argument for this set of simulations.

**Comments**

1. The title highlights KdV solutions with no mention of the eKdV solutions. I think it is well established by now that cubic nonlinearity is necessary to adequately model many observed solitary waves in the ocean, so if anything the eKdV equation should be mentioned in the title. Indeed, one wonders what the benefit of even considering the KdV equation is. Comparisons of the predictions of the KdV and eKdV (or Gardner) equations, as well as the RLW equation, with fully-nonlinear numerical simulations for a two-layer stratification are discussed in Lamb and Xiao (Ocean Modelling, 2014). This seems like a relevant reference.

2. Why are rotational effects not considered? The site of the observations is at mid-latitutude where rotation is going to affect the evolution of the internal tide and the amount of energy that ultimately gets transferred to ISWs. For example in Figure 10 the linear long wave propagation speed is about 0.5 m/s so waves take about

50 hours to travel 100 km. That is lots of time for rotation to affect their evolution.

3. First paragraph of page 4. Nonlinear effects can become important even without shoaling, as illustrated by the authors own flat-bottomed simulations so this should be reworded.

4. Page 5, lines 6–8. "It was originally developed in the context of internal waves by Benney ...."

5. Equations 10(b) and 10(c) are both incorrect.

6. Page 10. The introduction to section 3 repeats material from the introduction so should be deleted.

7. Page 12, line 15: For a given water depth *and wave amplitude* cubic ...". Then on lines 16–18, whether or not the eKdV model is similar to the KdV model depends a lot on the wave amplitude. For a two layer stratification, whenever the interface gets displaced close to the mid-depth cubic nonlinearity becomes important (though if $h_1/h_2 \ll 1$ higher-order nonlinear may be needed).

8. Section 3.1.1. The cases explored in this section are not well explained. All four cases have different total depths $H = h_1 + h_2$ and different depth ratios $h_1/h_2$ while from what I can understand the initial wave amplitude is the same in all cases. So both the the depth ratio $h_1/h_2$ and the initial nonlinearity have been changed. Comparing these cases is then a bit problematic, particularly with statements to the effect that you expect one case to be more nonlinear than the other. Also, throughout ratios such as $\alpha/c$ and $\beta/c$ are given without units. These ratios are not dimensionless. The KdV and eKdV equation have been used a lot to model internal solitary waves in the ocean. What have we learned from this set of simulations?

[Figure]

9. Section 3.1.2. What is new here? The general picture of the evolution of a shoaling internal tide has already been well described. What is the new contribution from this section?

10. Page 17, Line 5: there are no higher-order terms to prevent the development of solitary waves in the models used here.

11. Page 18, lines 14–15: What do you mean by 'We expect the waves to become unstable"? Do you mean your numerical solution is unstable? If so should a smaller time step be used? If a physical instability what type of instability is referred to?

12. Page 19, line 14: Do you mean the CMO line will be horizontal — lots of straight lines don't have constant $h_1$.

13. Page 21, lines 9–10. $\alpha/\alpha_1$ is not a dimensionless parameter.

14. Page 22, lines 3–8. Why is $\alpha_1$ so much greater at the CMO site than in case A? Is it because $h_1$ is so much less?

15. Page 25, ine 15. Do you mean figure 16c?

16. Page 26, 2nd paragraph. Something else that could be going on is the nonlinear evolution of inertia-gravity waves that form behind internal solitary waves due to rotation. See Grimshaw et al, JPO, 2014 or Lamb and Warn-Varnas, NPG, 2015. What about multiple packets forming each tidal period because of different generation mechanisms or multiple tidal constituents?

17. Page 29, lines 8–9. The internal tide is nonlinear right from the beginning — it doesn't become nonlinear sooner as $\beta$ is reduced. As $\beta$ is reduced waves have to get narrower before dispersive effects become significant.

18. Page 29, line 16. I think you mean if this ratio is much larger than one.

19. Page 30, line 8. What do you mean by 'the internal tide was forced with a sech$^2$ wave. Don't you mean the simulation was initialized with a sech$^2$ wave?

20. Figures. In general I find the font size too small in most of the figures - it is difficult to read them. In the caption for Figure 7 panels (a), (b) and (c) referred to in the text are not labeled. Figure 11 is of particularly poor quality.

---

## Author Comment (AC1) · 28 Feb 2017

Dear Handling Topic Editor,

I would like to thank the two reviewers for their reports and recommendations. I will revise the article accordingly and submit along with comments in due course.

Thank you,

Kieran O'Driscoll

---

## Author Response (AR1)

Dear John,

You will see that I have responded to all the Reviewers' comments, below, and edited accordingly.

Additionally, because of removal of the old sections 2. Theoretical Background, I have provided a little bit of background in the new section 2 for the non-expert. Concerning the removal of the old section 3.1.1 Level Bottom, all references to this have been removed throughout the paper: in the discussion of both KdV and eKdV model results, the summary and conclusions, and a very slight change in the abstract.

Thanks and best wishes,

Kieran O'Driscoll

Response to Referee #1 comments

***Kieran O'Driscoll would like to thank the reviewer for their considerable and thoughtful review.***

Based on the general comments of Reviewer 1 (and also those of Reviewer 2):

Much of the material presented here in section 2 and 3 is well-known and does not need to be repeated here.

Likewise, the behaviour of internal solitary waves propagating over a slope has been widely studied in both the KdV and eKdV models, with often a main focus on polarity change when the coefficient of the quadratic nonlinear term changes sign.

The simulations with a linear bottom slope and with the topography of the Middle Atlantic Bight have some marginal interest in that most studies have examined the behaviour of a single solitary wave, rather than a developing wave train as here, although the outcome can be understood in terms of the known behaviour of a single solitary wave, namely adiabatic deformation and transition to an elevation solitary wave train riding on a negative pedestal when the usual transition is from a negative to positive coefficient.

The most interesting and novel part of the paper is section 4 where the model simulations are compared with observational data from the Middle Atlantic Bight. Although there have been several such comparisons in the literature for other sites, this would seem to be the first for this site. In summary I would recommend that the authors prepare a heavily revised and shortened paper which focusses on the material in section 4.

*Response: Done. The article has been shortened by removing Sections 2 (Theoretical Background), 3.1.1. (Two-layer model level bottom), and Figs. 1 – 6.*

Specific Comments:

(1) Further to the comments made above, in particular most of the text in section is not needed, and neither are figures 1-6.
*Done, see above.*

(2) The measure χ (10a) of the relative roles of nonlinearity and dispersion is unconventional and uninformative. A better measure is simply the ratio $\alpha/\beta$, where it should be noted that in the KdV equation (1) division by $\beta$ and a rescaling of time, clearly indicates that this is the effective measure of nonlinearity vis-a-vis dispersion.

*Done. Removed with Section 2.*

(3) The transition of the steepening front into a solitary wave train is best understood using the Whitham modulation theory and asymptotic solution, as developed by Gurevich and Pitaevskii. It is well known that, at least in the KdV model, the leading waves are solitary waves. The detailed discussion on this aspect is not needed here.

*Done. The detailed discussion on this aspect has been removed with the old section 3.1.1 (previous version)*

(4) The large-amplitude solitary wave solutions of the eKdV equation are more usually called "table-top" waves than the term "tanh" used in the text.

*Done.*

(5) The title should mention "internal" and should not use an acronym.
*Done.*

Response to Referee # 2 comments

*Kieran O'Driscoll would like to thank the reviewer for their substantial and considerate review.*

Based on the general comments of Reviewer 2 (and also those of Reviewer 1):

I have a number of problems with the paper and think that is requires significant revision. My basic problem is that there doesn't appear to be much that is new here other than the application to the CMO site in the Middle Atlantic Bight and that is quite a small part of the paper.

The paper needs considerable polishing. Figures are in some cases hard to read and many dimensional values are given without units. It would probably benefit from being shortened and more focussed on the comparisons with observations however I am not convinced of the value of these simulations in that context. The authors make some comparisons of their results with those of Holloway et al from 30 years ago. Recent work has been done in this area using model equations that include rotation (e.g., Grimshaw and co-workers). The authors need to make a compelling argument for this set of simulations.

*Response: Done. The article has been shortened by removing Sections 2 (Theoretical Background), 3.1.1. (Two-layer model level bottom), and Figs. 1 – 6.*

**Comments**

1. The title highlights KdV solutions with no mention of the eKdV solutions. I think it is well established by now that cubic nonlinearity is necessary to adequately model many observed solitary waves in the ocean, so if anything the eKdV equation should be mentioned in the title. Indeed, one wonders what the benefit of even considering the KdV equation is. Comparisons of the predictions of the KdV and eKdV (or Gardner) equations, as well as the RLW equation, with fully-nonlinear numerical simulations for a two-layer stratification are discussed in Lamb and Xiao (Ocean Modelling, 2014). This seems like a relevant reference.

*Done, thanks.*

2. Why are rotational effects not considered? The site of the observations is at mid-latitutude where rotation is going to affect the evolution of the internal tide and the amount of energy that ultimately gets transferred to ISWs. For example in Figure 10 the linear long wave propagation speed is about 0.5 m/s so waves take about 50 hours to travel 100 km. That is lots of time for rotation to affect their evolution.

***The model is two-dimensional, so the waves propagate in the horizontal x-direction only and rotation is not included. This is stated in the abstract, discussion and summary.***

3. First paragraph of page 4. Nonlinear effects can become important even without shoaling, as illustrated by the authors own flat-bottomed simulations so this should be reworded.

***Done: Section 2 (Theoretical Background) has been removed***

4. Page 5, lines 6–8. "It was originally developed in the context of internal waves by Benney ...."

***Done: Section 2 (Theoretical Background) has been removed***

5. Equations 10(b) and 10(c) are both incorrect.

***Done: Section 2 (Theoretical Background) has been removed***

6. Page 10. The introduction to section 3 repeats material from the introduction so should be deleted.

***Done***

7. Page 12, line 15: For a given water depth and wave amplitude cubic ...". Then on lines 16–18, whether or not the eKdV model is similar to the KdV model depends a lot on the wave amplitude. For a two layer stratification, whenever the interface gets displaced close to the mid-depth cubic nonlinearity becomes important (though if $h_1/h_2 \ll 1$ higher-order nonlinear may be needed).

***Done: Section 3.1.1 (Two-layer model level bottom) has been removed***

8. Section 3.1.1. The cases explored in this section are not well explained. All four cases have different total depths $H = h_1 + h_2$ and different depth ratios $h_1/h_2$ while from what I can understand the initial wave amplitude is the same in all cases. So both the depth ratio $h_1/h_2$ and the initial nonlinearity have been changed. Comparing these cases is then a bit problematic, particularly with statements to the effect that you expect one case to be more nonlinear than the other. Also, throughout ratios such as $\alpha/c$ and $\beta/c$ are given without units. These ratios are not dimensionless. The KdV and eKdV equation have been used a lot to model internal solitary waves in the ocean. What have we learned from this set of simulations?

***Done: Section 3.1.1 (Two-layer model level bottom) has been removed***

9. Section 3.1.2. What is new here? The general picture of the evolution of a shoaling internal tide has already been well described. What is the new contribution from this section?

*These simulations studied the development of evolving internal tide as a packet or developing wave train across the linear sloping bottom, whereas most other studies have inspected the development and advance of a single soliton across similar bottom slope.*

10. Page 17, Line 5: there are no higher-order terms to prevent the development of solitary waves in the models used here.

*Sentence has been removed.*

11. Page 18, lines 14–15: What do you mean by 'We expect the waves to become unstable"? Do you mean your numerical solution is unstable? If so should a smaller time step be used? If a physical instability what type of instability is referred to?

*No, physically unstable, ie., Kelvin-Helmholtz instability or billows.*

*Done, thanks.*

12. Page 19, line 14: Do you mean the CMO line will be horizontal — lots of straight lines don't have constant h1.

*Yes, thanks. I will add Fig S1 (to replace old Fig 2) which shows values of h1, h2 for CMO.*

*Note: No need to add Fig S1 for this case, CMO parameter values are shown in Fig. 4a*

13. Page 21, lines 9–10. α/α1 is not a dimensionless parameter.

*Done, thanks. That was a typo, as seen from line 11, one line along. This section has now been removed.*

14. Page 22, lines 3–8. Why is α1 so much greater at the CMO site than in case A? Is it because h1 is so much less?

*Yes, thanks. Due to h1 half the value of h2. I have included Fig. S1 (α/α1) to show this. I have also included the equation:* $\frac{\alpha}{\alpha_1} = 4 \frac{h_1 - h_2}{h_1 h_2 \left(h_1{}^2 + h_2{}^2 + 6 h_1 h_2\right)}$

*while adding: ,* where $h_1$ and $h_2$ are, respectively, twice and less than that at the CMO site. Done.

15. Page 25, line 15. Do you mean figure 16c?

*Yes, thanks.*

16. Page 26, 2nd paragraph. Something else that could be going on is the nonlinear evolution of inertia-gravity waves that form behind internal solitary waves due to rotation. See Grimshaw et al, JPO, 2014 or Lamb and Warn-Varnas, NPG, 2015. What about multiple packets forming each tidal period because of different generation mechanisms or multiple tidal constituents?

*Thanks, done. References to these papers and alternate generation and evolution processes included as follows:*

Another possible generation mechanism is the nonlinear evolution of inertia-gravity waves forming behind internal solitary waves due to rotation, see further in Grimshaw et al. (2014) and Lamb & Warn-Varnas (2015). It is also possible that multiple packets form each tidal period, due to different generation mechanisms such as multiple tidal constituents or harmonics of a tidal components as found, for example, at the site of the Littoral Optics Experiment where the $4^{th}$ harmonic of the semi-diurnal tide was used to successfully simulate the evolution of the internal tide (O'Driscoll 1999).

17. Page 29, lines 8–9. The internal tide is nonlinear right from the beginning — it doesn't become nonlinear sooner as β is reduced. As β is reduced waves have to get narrower before dispersive effects become significant.

*Yes, thanks. Done. Changed accordingly. This has been removed since it is concerned with Cases 1-4, flat bottom and old Fig. 1-6.*

18. Page 29, line 16. I think you mean if this ratio is much larger than one

*Yes, thanks. Done. Edited accordingly.*

19. Page 30, line 8. What do you mean by 'the internal tide was forced with a sech2 wave. Don't you mean the simulation was initialized with a sech2 wave?

*No. It is forced, since it takes a tidal period for the wave to propagate into the model domain, i.e. the sech2 wave has tidal period.*

20. Figures. In general I find the font size too small in most of the figures – it is difficult to read them. In the caption for Figure 7 panels (a), (b) and (c) referred to in the text are not labeled. Figure 11 is of particularly poor quality

*Done. Figures with problem font sizes have been increased in size (because of removal of Figs. 1-6). Fig. 7 relabelled, Fig. 11 removed.*

[revised manuscript text omitted]

Fig.

|  |  |  |  |  |  |  |  |  |
|---|---|---|---|---|---|---|---|---|
|  |  |  |  |  |  |  |  |  |
|  |  |  |  |  |  |  |  |  |
|  |  |  |  |  |  |  |  |  |
|  |  |  |  |  |  |  |  |  |

5    S1. Quadratic nonlinear parameter, $\alpha$, divided by the cubic nonlinear parameter, $\alpha_1$, as a function of the depth of the upper layer, $h_1$, and lower layer, $h_2$. Values for sloping bottom (Cases A and B) and realistic slope and stratification at the CMO site are shown by the broken lines.

---

## Referee Report (RR1)

Review of "Simulations and observation of nonlinear waves on the continental shelf: KdV solutions" (revised) by *Kieran O'Driscoll and Murray Levine*

The paper has been significantly revised and greatly improved. I do not have any further major comments. But I can point the authors to a recent paper by Grimshaw and Yuan (2016) in Physica D volume 333, pp 200-207, who also integrated the KdV equation for propagation of undular bores up a slope. In particular that paper provides an explanation for the effect of a critical point of polarity change on an undular bore, as seen in the authors figure 1, panels 100, 110, 130 km, where after the critical point, one sees a depression rarefaction wave with elevation solitary waves riding on it. This leads to the lack of rank-ordering, and can be contrasted with their figure 3 where there is no polarity change, but is comparable with figure 4 where there is a polarity change.

---

## Referee Report (RR2)

*Review of*

**"Simulations and observation of nonlinear internal waves on the continental shelf: Korteweg-de Vries and extended Korteweg-de Vries solutions"**

*by K. O'Driscoll and M. Levine*

The author's justification for not considering rotational effects is not correct and is inadequate. A two-dimensional model just means the fields are functions of two spatial variables, say $x$ and $z$. There can still be flow in the third (lateral) direction. It is well known that rotation does affect internal solitary waves on time scales of the order of an inertial period. In particular energy is radiated behind the internal solitary wave in the form of long inertia-gravity waves. Given enough time the leading ISW can disappear and the trailing IG waves can steepen and form new ISWs. See, for example, Grimshaw et al (2014) or Lamb & Warn-Varnas (2015). In addition rotation can inhibit an internal tide from steepening and forming ISWs (e.g., Helfrich & Grimshaw 2008). So this begs the question of why the effects of rotation were not considered when making comparisons with the CMO data. Why isn't the Ostrovsky equation being used?

As a previous reviewer pointed out, refering to large amplitude ISW solutions of the eKdV equation as "tanh" waves is not recommended. There is an analytical expression for these waves in terms of tanh functions however the waves only look like a tanh function when they are very long with a flat-crest and none of the waves in this paper are remotely close to this shape. For this reason an alternative form of the solitary wave solutions, written in terms of a sech function, is usually used.

I think the paper needs considerable revision. The shoaling case with the sloping pycnocline is interesting and the comparisons with CMO observations is a useful addition to the literature however rotational effects should be included or a justification for not including them should be provided. An estimate could be made of the distance over which rotation would affect the waves.

1. Page 4, line 1. This sentence implies that the nonlinear terms are not important before the waves shoal which is not true

2. Line 5, line 16: "For the model cases ....." makes no sence

3. Page 5 equations 1 and 2. These equations don't fit in the sentence properly. Also, I suggest just giving the eKdV equation and saying that when $\alpha_1 = 0$ the equation reduced to the KdV equation.

4. Page 5, lines 20–21: "Nonlinear transformation of the internal tide leads to the generation of nonlinear waves". This doesn't make sense. You are saying a nonlinear wave leads to the generation of nonlinear waves. You mean

something like "Nonlinear steepening of the internal tide leads to the generation of a packet of short nonlinear waves which ...". On a similar vein, on page 8, lines 9 it is stated that 'the internal tide steepens and rapidly becomes nonlinear'. The internal tide is nonlinear from the start — otherwise it wouldn't be steepening! Following this what is meant by "the internal tide becomes more nonlinear"? What is the measure of the nonlinearity?

5. Page 6, lines 19–20. This sentence is not very clear. Perhaps "... we investigate two cases with a constant sloping bottom, one with a horizontal interface and one with a sloping interface, ..."

6. Page 7. Equation (4). Here a shoaling term that is not present in equations (1) and (2) is included. The first two equations should include the shoaling term. The shoaling term is not ever mentioned. I also suggest giving expressions for the nonlinear and dispersive parameters to readers can see, for example, how $\alpha$ changes sign, $\alpha_1 \neq 0$ and that $\beta > 0$.

7. Page 7. Please mention the numerical method used to solve the equations.

8. Page 7 line 15: "We chose *the* starting layer thickness*es* at $l = 0$ *to be ...* with *a* bottom ...'

9. Page 7, line 18: " ... at *a* water depth ..."

10. Page 9, line 3: Please define what is meant by the leading face? Since the internal tide is sinuosoidal this is ambiguous. Do you mean the front of the crest or the front of the trough?

11. Page 9, line 6: Do you mean the slope decreases rapidly? Above you say the leading face steepens but the steepening slows down, then you say the rate of change of the slope changes sign? I find this confusing.

12. Page 9, line 7. Where is the second shock-like front?

13. Page 9, lines 10–13. I don't see what the difference in the magnitudes of the nonlinear and dispersive terms has to do with the phase speed of the waves. Aren't waves propagating with phase speed less than $c$ simply because they are in the wave trough where $\eta < 0$? In the following paragraph mention of the wave alternating between being more nonlinear and dispersive is made. This does not make sense to me and this whole paragraph seems unhelpful. The nonlinear and dispersive terms vary across a given wave, changing sign at different locations in some instances. It does not seem helpful to say that at one value of $x$ where $\eta_{xxx} = 0$ and $\eta\eta_x \neq 0$ the wave is more nonlinear than

dispersive. The wave should be viewed as a whole. I trecommend removing this whole discussion along with Figures 2(a) and 4 and accompanying discussion.

14. Page 10, lines 5: The KdV or eKdV equation are not subject to Kelvin-Helmholtz instabilities.

15. Page 11, line 10: what is the 'back-face' of the wave? The back of the crest or the back of the trough?

16. Page 11, lines 12–13. "the number of solitary-like waves seems to have been reduced to the leading two waves". On what basis were some waves judged to be solitary-like and other waves not?

17. Page 11, lines 19–20: The instability must have something to do with the numerical scheme. The dispersion parameter may get small. It is never 0.

18. Page 12, lines 4. Here $\chi$ is defined. If use of $\chi$ is retained it should be defined when first used.

19. It would be useful to plot values of $\alpha_1$ when solutions of the eKdV equation are being discussed.

20. Page 19, line 9: When you say 'using CMO site parameters' are you using a continuous stratification? How do the parameters for the observed (i.e., continuous) stratification compare with those for the two-layer approximation? I am not convinced that a two-layer approximation is appropriate even if they waves are mode-one. A figure showing the observed stratification should be provided and justification for the choice $h_1 = 25$ m should be given.

21. Figures: The figures should be improved. For example in Figure 1(a) the ticks along the x-axis should be the same in all panels like they are in figure 2(a) (and I think ticks at 20 km intervals is better than at 30 km intervals). Many figures are of poor quality but perhaps I got low resolution versions of them? In Figure 7 it is hard to distinguish the two curves in the upper left panel. In this figure intervals of 11 along the bottom axis is odd. How about going by 10?

---

## Referee Report (RR3)

*Review of*
**"Simulations and observation of nonlinear internal waves on the continental shelf: Korteweg-de Vries and extended Korteweg-de Vries solutions"**
*by K. O'Driscoll and M. Levine*

1. I appreciate that rotational effects may not be large but after the times cited I would expect them to be apparent. See for example Figure 15(a) in Lamb and Warn-Varnas which shows differences after 23 hours at a latitude of about $20°$ N where rotational effects would be smaller than at the CMO site which is at $40°$ north. Figures 4 and 5 in Grimshaw *et al.* (JPO, 2015) also show differences by this time. Both of these papers are already cited. I do suggest a brief discussion of this be added.

2. Regarding 'solitary waves always travel faster than gravity waves, $V = c + \frac{\alpha \eta_0}{3}$. This is true for solitary waves propagating in an undisturbed medium but for solitary waves superimposed on other longer waves, e.g., and internal tide, they can propagate at less than $c$. For example if a solitary wave of depression with amplitude $\eta_{ISW} <$ is riding on a wave of elevation with amplitude $\eta_{elev} > 0$ sufficiently large. Consider the KdV equation with $\eta \to \eta_{elev} + \eta'$ where $\eta_{elev}$ is treated as a constant. Then $\eta'$ satisfies the KdV equation with that same nonlinear coefficient but $c$ replaced by $c + \alpha \eta_{elev} < c$.

3. Page 2, line 15: "Comparisons .... solutions *are made*."

4. Page 4, line 1. I would say it is the nonlinearity of fluid flow that causes the tidal waves to defore. Not the nonlinear terms in an equation.

5. Page 5, line 16: Do you mean $Q$ accounts for the horizontal variability of the ocean depth?

6. Page 5, lines 17: $M_0$ and $c_0$ are not defined.

7. Page 8, line 11: "layer *thicknesses* at ..."

8. Page 9, line 10: This needs rewording. Nonlinear waves are not prevented from developing into solitary waves because higher-order terms become of $O(\alpha)$ because here solutions of the KdV equation are being discussed and this equation has no higher-order terms to prevent the development of solitary waves.

9. Page 10, line 13: I suggest "so we expect a wave train to develop sooner ...." as the internal tide is nonlinear from the start.

10. Page 10, line 15: "... dispersive KdV *equation* becoming ..."

11. The author did change a number of statements like 'the leading face steepens' without stating whether the leading face is the leading side of the crest or trough but there are a few places where this change wasn't made: page 17, line 5; page 20k line 12; page 21, line 18

12. Page 12, last line: I am not sure why this figure is Figure S1. Supplementary material? I would keep it in the main body of the article. It is only one figure.

13. Page 11, line 20: I don't see why the first 4–5 waves look like solitary waves and the rest like a dispersive packet. Should explain this.

---

## Author Response (AR2)

[revised manuscript text omitted]

Dear John,

You will see that I have responded to all the Reviewers' comments, below, and edited the manuscript accordingly. A marked up version is also included (separate file).

Thanks and best wishes,

Kieran O'Driscoll

Response to Referee #1 comment

***Kieran O'Driscoll thanks the reviewer for their 2nd review.***

Review of "Simulations and observation of nonlinear waves on the continental shelf: KdV solutions" (revised) by Kieran O'Driscoll and Murray Levine The paper has been significantly revised and greatly improved. I do not have any further major comments. But I can point the authors to a recent paper by Grimshaw and Yuan (2016) in Physica D volume 333, pp 200-207, who also integrated the KdV equation for propagation of undular bores up a slope. In particular that paper provides an explanation for the effect of a critical point of polarity change on an undular bore, as seen in the authors figure 1, panels 100, 110, 130 km, where after the critical point, one sees a depression rarefaction wave with elevation solitary waves riding on it. This leads to the lack of rank-ordering, and can be contrasted with their figure 3 where there is no polarity change, but is comparable with figure 4 where there is a polarity change.

***Response: The following sentences have been included in relation to this feature (last paragraph of 2.1.1):***

These leading nonlinear waves mature into rank ordered solitary waves by $65km$, a feature observed out to $95km$. Grimshaw & Yuan (2016) have recently shown that the rank ordering is retained in shoaling waters as long as the waves do not encounter a critical point of polarity change on the waves of depression, i.e. $\alpha < 0$ always. This was not the case for Case A (above, Fig. 1, and CMO, below, Fig. 4).

Response to Referee # 2 comments

***Kieran O'Driscoll thanks the reviewer for their substantial 2nd review. I have reworked the paper to include all the reviewer's comments, see below.***

The author's justification for not considering rotational effects is not correct and is inadequate. A two-dimensional model just means the fields are functions of two spatial variables, say x and z. There can still be flow in the third (lateral) direction. It is well known that rotation does affect internal solitary waves on time scales of the order of an inertial period. In particular energy is radiated behind the internal solitary wave in the form of long inertia-gravity waves. Given enough time the leading ISW can disappear and the trailing IG waves can steepen and form new ISWs. See, for example, Grimshaw et al (2014) or Lamb & Warn-Varnas (2015). In addition, rotation can inhibit an internal tide from steepening and forming ISWs (e.g., Helfrich & Grimshaw 2008). So this begs the question of why the effects of rotation were not considered when making comparisons with the CMO data. Why isn't the Ostrovsky equation being used? As a previous reviewer pointed out, refering to large amplitude ISW solutions of the eKdV equation as "tanh" waves is not recommended. There is an analytical expression for these waves in terms of tanh functions however the waves only look like a tanh function when they are very long with a flat-crest and none of the waves in this paper are remotely close to this shape. For this reason an alternative form of the solitary wave solutions, written in terms of a sech function, is usually used. I think the paper needs considerable revision. The shoaling case with the sloping pycnocline is interesting and the comparisons with CMO observations is a useful addition to the literature however rotational effects should be included or a justification for not including them should be provided. An estimate could be made of the distance over which rotation would affect the waves.

***Thanks.*** *Grimshaw & Helfrich (2014) and Alias, Grimshaw & Khusnutdinova (2014) state that rotation effects become important after several inertial periods. For the 3 cases presented, Cases A, B and CMO, values of c are shown as the waves propagate shoreward in Figs. 1a, 3a 4a, respectively.*

*For Case A, for the first 100km of travel, solitary wave speed is estimated at*

$$V = c + \frac{\alpha \eta_0}{3} = 0.7 \text{m/s}.$$ *Given an inertial period of 18.67 hrs at 40N, the waves will travel about 95km in 2 inertial periods and into 130 km in a third.*

*For Case B, the waves will propagate the 95km considered in about 2 inertial periods.*

*For the CMO case, the waves are a bit slower and so rotation may become important beyond ~100km of travel. However, the discussion of results of KdV versus eKdV models (section 2.2) is mostly concerned with model runs to 60km (e.g. Fig. 6), as is the discussion comparing model results to observations (Figs. 9-13).*

*We did not include rotation in our model since it is based on the work and model of Holloway et al. (1997) who explicitly ignored rotation. Part of the reason we conducted this work was to compare model result waves with both theoretical solitons and waves observed waves at the CMO. And it has been discussed that exact solitary waves do not exist in the ocean due to effects of rotation, viscosity, etc., see e.g. Lamb & Xiao (2014).*

*All references to 'tanh' waves have been removed except in one instance, where the term 'table-top' is introduced and reads as follows p13, line 18:*

*Solitary type solutions to the KdV (sech$^2$) and to the eKdV ('table-top' waves with tanh type solutions, see e.g. O'Driscoll 1999 and references within) are fitted to the leading waves (Fig. 6d-e).*

**Comments**

1. Page 4, line 1. This sentence implies that the nonlinear terms are not important before the waves shoal which is not true

***Agreed, done. Thanks. Sentence has been changed to the following:***

As the internal tide shoals, the nonlinear terms in the Navier-Stokes equations can cause these tidal waves of finite amplitude to evolve into packets of high frequency nonlinear waves.

2. Page 4, line 16: "For the model cases ....." makes no sence

***Thanks, done. New sentence reads***

Whereas most modelling studies regarding wave propagation over linear bottom sloping realistic topography have focused on the behaviour of a single soliton, this work is concerned with the development and evolution of a packet of solitary waves.

3. Page 5 equations 1 and 2. These equations don't fit in the sentence properly. Also, I suggest just giving the eKdV equation and saying that when α1 = 0 the equation reduced to the KdV equation.

***Done. This now reads***

All cases have been run within the quadratic nonlinear framework of the KdV equation, and the results are compared with an extended form of it, the eKdV Eq.(1) model, written as

$$\eta_t + c\eta_x + (\alpha + \alpha_1\eta)\eta\eta_x + \beta\eta_{xxx} = 0 \qquad\qquad 1$$

which reduces to the KdV equation when $\alpha_1 = 0$.

For the KdV and eKdV equations to be valid, the leading two terms must constitute the primary balance ....

4. Page 5, lines 20–21: "Nonlinear transformation of the internal tide leads to the generation of nonlinear waves". This doesn't make sense. You are saying a

nonlinear wave leads to the generation of nonlinear waves. You mean something like "Nonlinear steepening of the internal tide leads to the generation of a packet of short nonlinear waves which ...". On a similar vein, on page 8, lines 9 it is stated that 'the internal tide steepens and rapidly becomes nonlinear'. The internal tide is nonlinear from the start — otherwise it wouldn't be steepening! Following this what is meant by "the internal tide becomes more nonlinear"? What is the measure of the nonlinearity?

***Thanks. Done. These sentences now read***

Nonlinear steepening of the internal tide leads to the generation of a packet of short nonlinear waves which tend to become solitary-like in form as the dispersive term becomes important. (p5, lines 19-21)

***p8. Yes, thanks. The input at the open boundary (sine wave) is linear but the system (model) is non-linear, so steepens as soon as it enters the model domain. The steepening is a measure of the nonlinearity, governed by the KdV/eKdV systems. So the sentence now reads***

The internal tide steepens from a sinusoid at the open boundary, rapidly becoming nonlinear, resulting in the generation of a shock-like front and subsequent undulations by $l \approx 50 km$. (p8, lines 10-12)

5. Page 6, lines 19–20. This sentence is not very clear. Perhaps "... we investigate two cases with a constant sloping bottom, one with a horizontal interface and one with a sloping interface, ..."

***Thanks. Done.***

6. Page 7. Equation (4). Here a shoaling term that is not present in equations (1) and (2) is included. The first two equations should include the shoaling term. The shoaling term is not ever mentioned. I also suggest giving expressions for the nonlinear and dispersive parameters to readers can see, for example, how α changes sign, α1 6= 0 and that β > 0.

***Thanks, done, as follows***

The term in $Q$ accounts for the horizontal variability of the ocean (see, e.g., Zhou & Grimshaw (1989), Pelinovsky et al. (1977) and Holloway et al. (1997)) and is given by

$Q = \dfrac{Mc^3}{M_0 c_0^{\,3}}$ with $M = \dfrac{h_1 + h_2}{h_1 h_2}$ where $h_1$ and $h_2$ are upper and lower layer thicknesses,

respectively. The coefficients of the KdV and eKdV equations are greatly simplified for a

two-layer fluid and are written (e.g. Ostrovsky & Stepanyants, 1989)  $c = \sqrt{\dfrac{g\Delta\rho}{\rho} \dfrac{h_1 h_2}{h_1 + h_2}}$ ;

$$\alpha = \frac{3}{2} c \frac{h_1 - h_2}{h_1 h_2} \; ; \qquad \beta = c \frac{h_1 h_2}{6} \; ; \qquad \alpha_1 = -\frac{3c}{8 h_1^{\,2} h_2^{\,2}} \left( h_1^{\,2} + h_2^{\,2} + 6 h_1 h_2 \right).$$

7. Page 7. Please mention the numerical method used to solve the equations.

***Done, reads:***

> We employ the same finite difference scheme as Holloway et al. (1997) to solve the eKdV equation Eq.(3) numerically. The finite difference scheme is a central difference method, (e.g. Lapidus & Pinder, 1982), which was first developed for the KdV equation by Berezin (1987), and for the variable coefficients KdV by Pelinovsky et al. (1994). See O'Driscoll (1999) for further details. (top p.8)

8. Page 7 line 15: "We chose the starting layer thicknesses at l = 0 to be ... with a bottom ...'

***Done***

9. Page 7, line 18: " ... at a water depth ..."

***Done.***

10. Page 9, line 3: Please define what is meant by the leading face? Since the internal tide is sinuosoidal this is ambiguous. Do you mean the front of the crest or the front of the trough?

***Done, thanks.***

the front of the wave trough

11. Page 9, line 6: Do you mean the slope decreases rapidly? Above you say the leading face steepens but the steepening slows down, then you say the rate of change of the slope changes sign? I find this confusing.

***Yes, thanks, done. I have just changed it to the slope decreases rapidly.***

12. Page 9, line 7. Where is the second shock-like front?

***Sentence removed.***

13. Page 9, lines 10–13. I don't see what the difference in the magnitudes of the nonlinear and dispersive terms has to do with the phase speed of the waves. Aren't waves propagating with phase speed less than c simply because they are in the wave trough where $\eta < 0$?

***No.*** Solitary waves always travel faster than gravity waves, $V = c + \dfrac{\alpha \eta_0}{3}$.

Given $\alpha = \dfrac{3}{2} c \dfrac{h_1 - h_2}{h_1 h_2}$ (these 2 expressions were included in the original version, see discussion paper), it is seen *α>0 when  $h_2 > h_1$*, thus, for waves of depression, *$\eta_0 < 0$*, so that α$\eta_0 > 0$, and *V>c.* The same argument is made for solitary waves of depression.
* * *
In the following paragraph mention of the wave alternating between being more nonlinear and dispersive is made. This does not make sense to me and this whole paragraph seems unhelpful. The nonlinear and dispersive terms vary across a given wave, changing sign at different locations in some instances. It does not seem helpful to say that at one value of x where ηxxx = 0 and ηηx 6= 0 the wave is more nonlinear than dispersive. The wave should be viewed as a whole. I recommend removing this whole discussion along with Figures 2(a) and 4 and accompanying discussion.

***Done. I removed reference to Fig 2b and Fig 5 and the associated lines.***

14. Page 10, lines 5: The KdV or eKdV equation are not subject to KelvinHelmholtz instabilities.

***Sentence removed.***

15. Page 11, line 10: what is the 'back-face' of the wave? The back of the crest or the back of the trough?

***Thanks. Same as 10 above:*** on the front of the internal tide wave trough

16. Page 11, lines 12–13. "the number of solitary-like waves seems to have been reduced to the leading two waves". On what basis were some waves judged to be solitary-like and other waves not?

***From Figs. 4b and c.*** *Solitary waves travel faster than regular gravity waves:*

$V = c + \dfrac{\alpha \eta_0}{3}$ (e.g. O'Driscoll 1999), included in original version (see OSD manuscript). So, the troughs of solitary waves (blue) travel to the left, Fig. 4c. Only the first two travel to the left after 80km.

17. Page 11, lines 19–20: The instability must have something to do with the numerical scheme. The dispersion parameter may get small. It is never 0

***Agreed. The sentence has been changed to the following:***

The internal tide becomes numerically unstable beyond $l = 130 km$.

18. Page 12, lines 4. Here χ is defined. If use of χ is retained it should be defined when first used

***Done. All reference to χ has been removed, see 13 above.***

19. It would be useful to plot values of α1 when solutions of the eKdV equation are being discussed.

***Sorry, this was included in the previous manuscript, Fig. S1, I forgot to reference it. Thanks.***

20. Page 19, line 9: When you say 'using CMO site parameters' are you using a continuous stratification? How do the parameters for the observed (i.e., continuous)

stratification compare with those for the two-layer approximation? I am not convinced that a two-layer approximation is appropriate even if they waves are mode-one. A figure showing the observed stratification should be provided and justification for the choice h1 = 25 m should be given.

*A 2-layer stratification was used, which was a realistic model given conditions at the CMO site in August 1996, and was chosen following discussed with my advisor, Murray Levine. Examples of the stratification at the CMO site are given in* Boyd, T., Levine, M.D., and Gard, S.R.: Mooring observations from the Mid-Atlantic Bight, Oregon State University, Data Report 97-2, pp. 226, 1997 (referenced in text). Also, please see below a section across the site of the CMO shelf, shows an upper mixed layer of ~25m separated from a very weakly stratified pycnocline (from Barth et al. 1998, referenced).

[Figure]

**Figure 1.** Cross-shelf sections of temperature, salinity and density along 70.48°W from 13:01 to 19:35 UTC on August 21, 1996. Gridded data locations are shown as light dots. The location of the cross-shelf section is indicated by a thick line on the map.

21. Figures: The figures should be improved. For example in Figure 1(a) the ticks along the x-axis should be the same in all panels like they are in figure 2(a) (and I think ticks at 20 km intervals is better than at 30 km intervals). Many figures are of poor quality but perhaps I got low resolution versions of them? In Figure 7 it is hard to distinguish the two curves in the upper left panel. In this figure intervals of 11 along the bottom axis is odd. How about going by 10?

*Thanks. Tick marks are the same for all panels Fig. 1, 3, 4.*

*The close up in Fig 7b is a close up of 7a (now Fig. 6).*

*We used 44000s as the semi-diurnal period, so 11000s is ¼ of the period, etc.*

---

## Author Response (AR3)

Response to Reviewer 2 (version 3)

Review of "Simulations and observation of nonlinear internal waves on the continental shelf: Korteweg-de Vries and extended Korteweg-de Vries solutions"
by K. O'Driscoll and M. Levine

Kieran O'Driscoll would like to thank the reviewer for their very thorough examination of the manuscripts.

2. I appreciate that rotational effects may not be large but after the times cited I would expect them to be apparent. See for example Figure 15(a) in Lamb and Warn-Varnas which shows differences after 23 hours at a latitude of about 20∘ N where rotational effects would be smaller than at the CMO site which is at 40∘ north. Figures 4 and 5 in Grimshaw et al. (JPO, 2015) also show differences by this time. Both of these papers are already cited. I do suggest a brief discussion of this be added.

***Thanks, done.*** *The following sentences have been included in the discussion on* **3.1  Observations during the Coastal Mixing and Optics Experiment**
*In the paragraph that starts* There are also features of the observations that are not found ….
*The sentence:* Another possible generation mechanism is the nonlinear evolution of inertia-gravity waves forming behind internal solitary waves due to rotation, see further in Grimshaw et al. (2014) and Lamb & Warn-Varnas (2015).
*Has been replaced with*:
Another possible generation mechanism is the nonlinear evolution of inertia-gravity waves forming behind internal solitary waves due to rotation, see further in Grimshaw et al. (2014) and Lamb & Warn-Varnas (2015) who have shown that rotation effects can become important after one or more inertial periods. However, in this model to observation comparison, the waves have travelled for not much longer than one inertial period, and rotation has been ignored in the model runs.

2. Regarding 'solitary waves always travel faster than gravity waves, $V = c + \dfrac{\alpha \eta_0}{3}$. This is true for solitary waves propagating in an undisturbed medium but for solitary waves superimposed on other longer waves, e.g., and internal tide, they can propagate at less than c. For example if a solitary wave of depression with amplitude $\eta_{ISW} <$ is riding on a wave of elevation with amplitude $\eta_{elev} > 0$ sufficiently large. Consider the KdV equation with $\eta \rightarrow \eta_{elev} + \eta'$ where $\eta_{elev}$ is treated as a constant. Then $\eta'$ satisfies the KdV equation with that same nonlinear coefficient but c replaced by $c + \alpha_{\eta elev} < c$.
***Thank you. Yes, I agree.***

3. Page 2, line 15: "Comparisons .... solutions are made."
***Done.***

4. Page 4, line 1. I would say it is the nonlinearity of fluid flow that causes the tidal waves to defore. Not the nonlinear terms in an equation.

***Done.*** *Sentence now reads:* As the internal tide shoals, the nonlinearity of fluid flow can cause these tidal waves of finite amplitude to evolve into packets of high frequency nonlinear waves.

5. Page 5, line 16: Do you mean Q accounts for the horizontal variability of the ocean depth?
***Yes, thanks. Done:*** The term in $Q$ accounts for the horizontal variability of the ocean depth

6. Page 5, lines 17: $M_0$ and $c_0$ are not defined.
***Done,*** *the sentence now reads* The term in $Q$ accounts for the horizontal variability of the ocean depth (see, e.g., Zhou & Grimshaw (1989), Pelinovsky et al. (1977) and Holloway et al. (1997)) and is given by $Q = \dfrac{Mc^3}{M_0 c_0^{\,3}}$ with $M = \dfrac{h_1 + h_2}{h_1 h_2}$ where $h_1$ and $h_2$ are upper and lower layer thicknesses, respectively, and the zero subscript indicates a constant value at a predetermined starting position.

7. Page 8, line 11: "layer thicknesses at ..."
***Done***

8. Page 9, line 10: This needs rewording. Nonlinear waves are not prevented from developing into solitary waves because higher-order terms become of O(α) because here solutions of the KdV equation are being discussed and this equation has no higher-order terms to prevent the development of solitary waves.

***Thank you, done.***
*The sentence:* However, as $\alpha \to 0$ the nonlinear waves are prevented from developing into solitary waves, since higher order terms (neglected in the KdV) become of order $\alpha$ or larger and thus cannot be ignored, thereby rendering the KdV model invalid in this neighborhood.
*has been replaced with:* Approaching $l = 100km$, $\alpha \to 0$ and the nonlinear waves cannot develop into solitary waves.
*And is followed by the sentence:* At $l = 100km$ the packet certainly looks symmetrical about a horizontal axis, that is to say the waves are neither polarized as waves of depression nor elevation, since KdV solitary waves cannot exist when $\alpha = 0$.

9. Page 10, line 13: I suggest "so we expect a wave train to develop sooner ...." as the internal tide is nonlinear from the start.

***Done***

10. Page 10, line 15: "... dispersive KdV equation becoming ..."
**Done**

11. The author did change a number of statements like 'the leading face steepens' without stating whether the leading face is the leading side of the crest or trough but there are a few places where this change wasn't made: page 17, line 5; page 20k line 12; page 21, line 18

**Done.** *These now read:* page 17, line 5 … the leading face of the crest of the periodic sinusoidal wave slackens …
page 20 line 12 …. The internal tide steepens on the back face of its crest as it propagates shoreward …
page 21, line 18 … The leading face of the trough of the internal tide steepens …

12. Page 12, last line: I am not sure why this figure is Figure S1. Supplementary material? I would keep it in the main body of the article. It is only one figure.

**Done.** *This is now Fig. 5 and all subsequent figure numbers have been changed accordingly*

13. Page 11, line 20: I don't see why the first 4–5 waves look like solitary waves and the rest like a dispersive packet. Should explain this

**Done***, the sentence has been amended and appended as follows:*
Several nonlinear waves have formed by $l = 60km$ (mooring location) with the leading $4 - 5$ waves appearing like solitary waves of depression and the trailing waves looking more like a dispersive packet, i.e., the leading waves travel faster than $c$, to the left for increasing $l$ in Fig. 4c.